# Adaptive Proximal Gradient Optimizer: Addressing Gradient Inexactness in Predict+Optimize Framework

## Abstract

To achieve end-to-end optimization in the Predict+Optimize (P+O) framework, efforts have been focused on constructing surrogate loss functions to replace the non-differentiable decision regret. While these surrogate functions are effective in forwarding training, the backpropagation of the gradient introduces a significant but unexplored problem: the inexactness of the surrogate gradient, which often destabilizes the training process. To address this challenge, we propose the Adaptive Proximal Gradient Optimizer (AProx), the first gradient descent optimizer designed to handle the inexactness of surrogate gradient backpropagation within the P+O framework. Instead of explicitly solving proximal operations, AProx uses subgradients to approximate the proximal operator, simplifying the computational complexity and making proximal gradient descent feasible within the P+O framework. We prove that the surrogate gradients of three major types of surrogate functions are subgradients, allowing efficient application of AProx to end-to-end optimization. Additionally, AProx introduces momentum and novel strategies for adaptive weight decay and parameter smoothing, which together enhance both training stability and convergence speed. Through experiments on several classical combinatorial optimization benchmarks using different surrogate functions, AProx demonstrates superior performance in stabilizing the training process and reducing the optimality gap under predicted parameters.

## 1 Introduction

End-to-end learning has demonstrated powerful representational capabilities in prediction tasks, driving revolutionary breakthroughs in computer vision (He et al. (2016)), and natural language processing (Vaswani (2017)). The end-to-end approach is an emerging approach that has the potential to change tradition in the decision-making process. For example, in autonomous driving, the UniAD proposes an integrated framework for end-to-end perceptual decision-making that coordinates perceptual predictive decision-making to enhance path-planning capabilities (Hu et al. (2023b)). In the areas of maternal and child health (Wang et al. (2023)), and environmental change (Harder et al. (2023)), research has been invested to enable end-to-end decision-making and maximize social value.

Most end-to-end decisions can be inseparable from the prediction and the optimization stage. In the prediction stage, the machine learning model generates predicted values for unknown parameters, which are subsequently fed to the optimization model. This two-stage approach will lead to the problem of error misalignment between the prediction and the optimization model. Therefore, less prediction error can not ensure a minor gap between the predicted and the true optimal value (Geng et al. (2023)).

To solve this problem, Predict+Optimize (P+O) is developed to integrate the two stages of prediction and optimization into one, enabling end-to-end training from features to predicted optimal values (Demirović et al. (2019)). Although the P+O approach is straightforward, the loss function representing decision regret is non-differentiable. Currently, one major solution is to construct a surrogate loss function, thus acquiring surrogate gradients as approximations for end-to-end training (Mulamba et al. (2021); Elmachtoub & Grigas (2022); Guler et al. (2022); Ferber et al. (2023)).

However, such surrogate gradients are inexact since error always exists in the approximation from the surrogate function to the ideal differentiable loss function.

The inexactness of surrogate gradients surely needs to be emphasized for its importance during the training process of the P+O framework. Existing optimizers used for end-to-end training are designed for exact gradients (Schaul et al. (2013); Kingma (2014)), lacking the consideration of inexact gradients in the P+O framework. The presence of inexact gradients often fails to provide a direction consistent with the steepest descent path. This mismatch leads to slower convergence and parameter update vibration, impacting the stability and efficiency of the training process.

Current research on the impact of inexact gradients in optimization mostly focuses on the solution process of optimization problems(Yang & Li (2023); Barré et al. (2023)). In end-to-end Predict+Optimize (P+O) frameworks, the problem of gradient inexactness during training remains under-explored. To bridge this gap, we propose the Adaptive Proximal Gradient Optimizer (AProx). Unlike traditional optimizers, AProx builds on proximal gradient descent and implicitly computes the proximal operator. By integrating momentum terms along with improved weight decay and parameter smoothing strategies, AProx effectively handles the inexactness of the surrogate gradient during backpropagation, improving the stability of training. Our contributions can be summarized as follows:

- We introduce the Adaptive Proximal Gradient Optimizer (AProx), the first to directly address the importance of inexact surrogate gradient backpropagation in end-to-end optimization. AProx employs an implicit proximal gradient descent method using subgradients, which is independent of the specific surrogate regret function. This characteristic makes it broadly applicable across various surrogate-based solutions of the P+O framework.
- We enhance the optimizer by incorporating momentum along with newly designed adaptive weight decay and parameter smoothing strategies. These enhancements improve the stability and efficiency of training, balancing convergence speed with robustness, especially in the presence of surrogate gradient inaccuracies.
- We theoretically prove that the surrogate gradients of three classes of surrogate functions for P+O are subgradients, which allows the effective use of AProx in these scenarios. Furthermore, we establish the convergence properties of AProx when applied to these surrogate gradients, demonstrating its effectiveness in stabilizing the training process and improving solution quality.

## 2 INEXACTNESS OF SURROGATE GRADIENTS IN P+O FRAMEWORK

### 2.1 SURROGATE GRADIENT REQUIREMENT

The Predict+Optimize (P+O) framework involves making end-to-end decisions based on predictions of uncertain parameters. Formally, consider a decision variable $z \in \mathbb{R}^n$, an input feature vector $x \in \mathbb{R}^d$, and true cost parameters $c \in \mathbb{R}^n$. The target is to learn a predictive model $\hat{c} = \phi(\theta)$ parameterized by $\theta$ that maps $x$ to estimated costs. Then solve an optimization problem to determine the optimal decision $z^*(\hat{c}) = \arg\min_{z \in \mathcal{Z}} \hat{c}^\top z$, where $\mathcal{Z}$ is the feasible set defined by problem-specific constraints.

To achieve end-to-end optimization, the P+O framework defines a decision regret function as:

$$R(\hat{c}) = c^\top z^*(\hat{c}) - c^\top z^*(c). \tag{1}$$

The key challenge lies in computing the gradient of the regret function for $\theta$. During the training process, the chain rule for differentiation would require the following:

$$\nabla_\theta R(\hat{c}) = \nabla_{\hat{c}} R(\hat{c}) \cdot \nabla_\theta \hat{c}(\theta) = \nabla_{\hat{c}} \left( c^\top z^*(\hat{c}) \right) \cdot \left( \nabla_\theta \hat{c}(\theta) \right)^\top. \tag{2}$$

In practice, the predicted cost vector $\hat{c}(\theta)$ is obtained from differentiable neural networks or machine learning models, making the computation of $\nabla_\theta \hat{c}(\theta)$ relatively straightforward. However, the primary challenge is that $z^*(\hat{c})$ comes from an optimization problem, which is often non-differentiable.

Current approaches mainly focus on constructing a surrogate function $\tilde{R}(\hat{c})$ to approximate the decison regret $R(\hat{c})$. Consequently, *surrogate gradients* are computed through surrogate functions, enabling gradient-based end-to-end optimization.

## 2.2 Challenge of Surrogate Gradient Inexactness

Although the difficulty of constructing surrogate functions has been addressed through different approaches, the effects of end-to-end training with surrogate gradients are not sufficiently discussed. In these settings, accurate gradient computation is crucial for the convergence and stability of the training process.

However, when using surrogate functions, the gradients are computed inexactly: $\tilde{g}(\hat{c}) \approx \nabla_{\hat{c}} \left( c^\top z^*(\hat{c}) \right)$. Such inexactness can introduce an error compared with the ideal true gradient:

$$\left\| \tilde{g}(\hat{c}) - \nabla_{\hat{c}} \left( c^\top z^*(\hat{c}) \right) \right\| \le \delta, \tag{3}$$

where $\delta > 0$ represents the error bound, which is often non-negligible.

While backpropagating through the equation (2), the inexactness will be accumulated:

$$\|\epsilon_t\| \le \delta \left\| \nabla_{\theta} \hat{c}(\theta_t) \right\|$$

where $\epsilon_t = \nabla_{\theta} \tilde{R}(\hat{c}_t) - \nabla_{\theta} R(\hat{c}_t)$ and . This leads to instability in the training process, making parameter updates unstable and convergence difficult in end-to-end training.

Therefore, reducing the impact of surrogate gradient inexactness is crucial to ensure stable training of the P+O framework. This requires not only improving the accuracy of the surrogate gradient, but also developing new optimizers that enhance the backpropagation process to adapt to the inexact surrogate gradient.

# 3 Adaptive Proximal Gradient Optimizer (APROX)

In this section, we will briefly introduce the core ideas behind the Adaptive Proximal Gradient Optimizer (AProx), a novel optimizer specifically designed to address the inexact gradient challenges within P+O framework. Most of the lemmas and corresponding proofs will be detailed in later sections.

## 3.1 Implicit Proximal Gradient Descent

A key innovation of AProx is the introduction of proximal gradient descent instead of standard gradient descent. This allows AProx to inherently accommodate the inexactness of surrogate gradients within the basic update rules, providing better robustness and adaptability to such inaccuracies.

To effectively address the inexact gradient issue, we first construct a composite function that incorporates the regret term $\tilde{R}(\hat{c})$ as follows:

$$F(\hat{c}) = f(\hat{c}) + \tilde{R}(\hat{c}), \tag{4}$$

where $f(\hat{c})$ is a smooth, differentiable loss function. Here, $f(\hat{c})$ is defined as $f(\hat{c}) = \frac{1}{2}|\hat{c} - c|^2$, where the factor $\frac{1}{2}$ helps to avoid redundant coefficients during differentiation and can also act as a weighting factor for regularization purposes. This function aids in minimizing decision regret while introducing a regularization effect. The non-smooth term $\tilde{R}(\hat{c})$ represents the approximated convex surrogate function, which we construct using three kinds of convex surrogate functions, Perturbed Methods (Niepert et al. (2021); Berthet et al. (2020); Minervini et al. (2023)), Contrastive Methods (Mulamba et al. (2021)), and Convex Upper Bound Methods (Elmachtoub & Grigas (2022)).

To handle the non-smooth nature of the surrogate function, we introduce the proximal gradient descent given by:

$$\hat{c}_{t+1} = \text{prox}_{\eta \tilde{R}} \left( \hat{c}_t - \eta \nabla f(\hat{c}_t) \right), \tag{5}$$

where $\eta > 0$ is the learning rate, and $\text{prox}_{\eta \tilde{R}}$ is the proximal operator associated with $\tilde{R}$, defined as:

$$\text{prox}_{\eta \tilde{R}}(v) = \arg\min_{\hat{c}} \left\{ \tilde{R}(\hat{c}) + \frac{1}{2\eta}|\hat{c} - v|^2 \right\}, \tag{6}$$

where $v$ is proximal point of $\hat{c}$. This approach provides a mechanism to update parameters while considering the non-smooth properties of the surrogate function.

To make the proximal gradient update more efficient for end-to-end learning, we employ an implicit method using subgradients, as shown in the following lemma.

**Lemma 1** (Implict Proximal Gradient Descent Update rule). *Let $\tilde{R} : \mathbb{R}^n \to \mathbb{R}$ be a convex, non-smooth function, and let $f : \mathbb{R}^n \to \mathbb{R}$ be a differentiable convex function with an L-Lipschitz continuous gradient. The proximal gradient descent update formula $\hat{\boldsymbol{c}}_{t+1} = \text{prox}_{\eta\tilde{R}}(\hat{\boldsymbol{c}}_t - \eta\nabla f(\hat{\boldsymbol{c}}_t))$ can be equivalently written in the form:*

$$\hat{\boldsymbol{c}}_{t+1} = \hat{\boldsymbol{c}}_t - \eta\nabla f(\hat{\boldsymbol{c}}_t) - \eta\tilde{g}_t,$$

*where $\tilde{g}_t \in \partial\tilde{R}(\hat{\boldsymbol{c}}_{t+1})$ represents a subgradient of $\tilde{R}$ at $\hat{\boldsymbol{c}}_{t+1}$.*

For convenience of presentation, we will use $\boldsymbol{g}_t$ to replace $\nabla f(\hat{\boldsymbol{c}}_t) + \tilde{g}_t$. This implicit proximal gradient descent update rule allows us to incorporate the subgradient of the surrogate function, making the update computationally efficient and suitable for end-to-end learning scenarios. In this context, $\tilde{\boldsymbol{g}}_t$ essentially represents the surrogate gradient, and the corresponding proof will be provided in a later section.

### 3.2 APROX OPTIMIZER STRATEGIES

In the AProx optimizer, we incorporate several strategies based on the implicit proximal gradient update rule. These strategies are utilized to improve stability, convergence speed, and generalization, some of which have been shown to be effective in gradient-based optimization.

**Incorporating First-order Momentum (Kingma (2014))**: Momentum can average the noise in inexact gradients over time, leading to a more reliable search direction. Therefore, we apply the first-order moment estimate $\boldsymbol{m}_t$ as:

$$\boldsymbol{m}_t = \beta_1\boldsymbol{m}_{t-1} + (1 - \beta_1)\boldsymbol{g}_t,$$

where $\beta_1 \in [0, 1)$ is the momentum coefficient. The use of momentum allows us to smooth the sequence of gradient estimates over time, improving the stability of updates.

**Incorporating Adaptive Learning Rates**: For adaptive learning rates, we compute a biased second-order momentum estimate:

$$\boldsymbol{v}_t = \beta_2\boldsymbol{v}_{t-1} + (1 - \beta_2)\boldsymbol{g}_t^2,$$

where $\beta_2 \in [0, 1)$ is the decay rate for the second-order momentum estimate. The maximum correction (Loshchilov & Hutter (2019)) is then used in AProx to ensure that the adaptive learning rate does not decay too quickly:

$$\hat{\boldsymbol{v}}_t = \max(\hat{\boldsymbol{v}}_{t-1}, \boldsymbol{v}_t).$$

This correction addresses convergence issues by preventing a vanishing learning rate. The bias-corrected first-order momentum estimate is given by:

$$\hat{\boldsymbol{m}}_t = \frac{\boldsymbol{m}_t}{1 - \beta_1^t}.$$

This bias correction mitigates the initial underestimation of the first-order momentum, particularly at the early stages of training when the accumulated gradient information is limited.

Using the corrected first-order momentum and corrected second-order momentum, we compute the adaptive learning rate for each parameter:

$$\boldsymbol{\eta}_t = \alpha\frac{1}{\sqrt{\hat{\boldsymbol{v}}_t} + \epsilon},$$

where $\alpha > 0$ is the base learning rate and $\epsilon > 0$ is a small constant for numerical stability. This adaptive adjustment enables the optimizer to scale the learning rate effectively, taking larger steps in directions with low variance and smaller steps where gradients are large or noisy (as shown in the challenge of surrogate gradient inexactness).

**Temporal Averaging for Parameter Robustness**: To further enhance the stability of the model and improve generalization, we maintain a running average of the model parameters:

$$\hat{\boldsymbol{c}}_{\text{avg},t} = \gamma\hat{\boldsymbol{c}}_{\text{avg},t-1} + (1 - \gamma)\hat{\boldsymbol{c}}_t,$$

where $\gamma \in [0, 1)$ is the parameter smoothing coefficient. Compared with existing optimizer strategies, the introduced strategy runs an average of model parameters. By averaging parameters over time, AProx is capable of reducing sensitivity to inexact surrogate gradient updates.

---

**Algorithm 1** Adaptive Proximal Gradient Optimizer (AProx)

---

**Require:** Initial moments $\boldsymbol{m}_0 = \boldsymbol{0}$, $\boldsymbol{v}_0 = \boldsymbol{0}$, initial parameter average $\hat{\boldsymbol{c}}_{\text{avg},0} = \hat{\boldsymbol{c}}_0$, hyperparameters $\alpha > 0$, $\beta_1 \in [0, 1)$, $\beta_2 \in [0, 1)$, $\gamma \in [0, 1)$, $\lambda \geq 0$, $\epsilon > 0$

1: **for** $t = 1$ to $T$ **do**
2:     Compute gradient of smooth loss function: $\nabla f(\hat{\boldsymbol{c}}_t)$
3:     Compute subgradient of surrogate function: $\tilde{\boldsymbol{g}}_t \in \partial \tilde{R}(\hat{\boldsymbol{c}}_t)$
4:     Compute total gradient: $\boldsymbol{g}_t = \nabla f(\hat{\boldsymbol{c}}_t) + \tilde{\boldsymbol{g}}_t$
5:     Update biased first-order momentum estimate: $\boldsymbol{m}_t = \beta_1 \boldsymbol{m}_{t-1} + (1 - \beta_1)\boldsymbol{g}_t$
6:     Update biased second-order momentum estimate: $\boldsymbol{v}_t = \beta_2 \boldsymbol{v}_{t-1} + (1 - \beta_2)\boldsymbol{g}_t^2$
7:     Apply maximum correction: $\hat{\boldsymbol{v}}_t = \max\left(\hat{\boldsymbol{v}}_{t-1}, \boldsymbol{v}_t\right)$
8:     Compute bias-corrected first moment estimate: $\hat{\boldsymbol{m}}_t = \dfrac{\boldsymbol{m}_t}{1 - \beta_1^t}$
9:     Compute adaptive learning rate: $\boldsymbol{\eta}_t = \alpha \dfrac{1}{\sqrt{\hat{\boldsymbol{v}}_t} + \epsilon}$
10:     **if** weight decay $\lambda > 0$ **then**
11:         Update parameters with adaptive weight decay: $\hat{\boldsymbol{c}}_{t+1} = \dfrac{1}{1 + \alpha\lambda}\left(\hat{\boldsymbol{c}}_{\text{avg},t} - \boldsymbol{\eta}_t \odot \hat{\boldsymbol{m}}_t\right)$
12:     **end if**
13:     Update parameter average: $\hat{\boldsymbol{c}}_{\text{avg},t} = \gamma\hat{\boldsymbol{c}}_{\text{avg},t-1} + (1 - \gamma)\hat{\boldsymbol{c}}_t$
14: **end for**
15: **Return** $\hat{\boldsymbol{c}}_{\text{avg},T}$ or $\hat{\boldsymbol{c}}_T$

---

**Adaptive Regularization via Weight Decay Dynamics**: To prevent overfitting and control model complexity, we introduce weight decay during the parameter update:

$$\hat{\boldsymbol{c}}_{t+1} = \frac{1}{1 + \alpha\lambda}\left(\hat{\boldsymbol{c}}_{\text{avg},t} - \boldsymbol{\eta}_t \odot \hat{\boldsymbol{m}}_t\right),$$

where $\lambda \geq 0$ is the weight decay coefficient. $\odot$ denotes the element-wise multiplication. Unlike traditional fixed weight decay methods used in optimizers like AdamW, AProx employs an adaptive approach to scale the parameter update. Specifically, the operator $\frac{1}{1+\alpha\lambda}$ is applied element-wise, meaning each parameter $\hat{\boldsymbol{c}}_{t,i}$ is individually adjusted according to the weight decay factor, which adapts based on the learning rate and the coefficient $\lambda$. This effectively reduces the influence of weight decay when the learning rate is lower, maintaining parameter stability.

By integrating these strategies, AProx can address the challenges of inexact surrogate gradients in the P+O framework. The detailed steps of the AProx algorithm are presented in Algorithm 1. Comparison with AProx and existing baseline optimizers is shown in section B.

## 4 THEORETICAL CONVERGENCE ANALYSIS OF APROX

In this section, we present the convergence analysis of the proposed Adaptive Proximal Gradient Optimizer (AProx). The detailed proof procedures for each of the following results are provided in the appendix.

First, we will give lemmas to illustrate the convexity and subgradient properties of the three classes of generating functions used in AProx. Each of the following Lemma is essential for verifying Lemma 2, as they prove both the convexity of a particular surrogate function and the validity of its surrogate gradient as a subgradient.

**Lemma 2** (Surrogate Gradient of Perturbed Methods IN P+O). *The perturbed surrogate loss function $L_{pert}(\mathbf{c}, \hat{\mathbf{c}})$ (Niepert et al. (2021)) is given by*

$$L_{pert}(\mathbf{c}, \hat{\mathbf{c}}) = \mathbb{E}_{\hat{\mathbf{z}} \sim q(\mathbf{z}; \hat{\mathbf{c}})}\left[A(\mathbf{c}) - \langle \hat{\mathbf{z}}, \mathbf{c} \rangle\right],$$

*is convex with respect to $\mathbf{c}$. Moreover, the surrogate gradient*

$$g_{pert} = \boldsymbol{\mu}(\mathbf{c}) - \boldsymbol{\mu}(\hat{\mathbf{c}}),$$

*where $\boldsymbol{\mu}(\mathbf{c}) = \nabla_{\mathbf{c}} A(\mathbf{c})$, is a subgradient of $L_{pert}(\mathbf{c}, \hat{\mathbf{c}})$ at $\mathbf{c}$.*

**Lemma 3** (Surrogate Gradient of Contrastive Methods IN P+O). *The CMAP surrogate loss function* $L_{contrast}(\hat{\mathbf{c}}, \mathbf{c})$ *(Mulamba et al. (2021)) is given by*

$$L_{contrast}(\hat{\mathbf{c}}, \mathbf{c}) = \frac{1}{|\Gamma| - 1} \sum_{\mathbf{z} \in \Gamma \backslash \{\mathbf{z}^*(\mathbf{c})\}} \left( \hat{\mathbf{c}}^\top \mathbf{z}^*(\mathbf{c}) - \hat{\mathbf{c}}^\top \mathbf{z} \right),$$

*is convex with respect to* $\hat{\mathbf{c}}$*. Moreover, the surrogate gradient*

$$g_{contrast} = \frac{1}{|\Gamma| - 1} \sum_{\mathbf{z} \in \Gamma \backslash \{\mathbf{z}^*(\mathbf{c})\}} (\mathbf{z}^*(\mathbf{c}) - \mathbf{z})$$

*is a subgradient of* $L_{contrast}(\hat{\mathbf{c}}, \mathbf{c})$ *at* $\hat{\mathbf{c}}$*.*

**Lemma 4** (Surrogate Gradient of Upper Bound Methods IN P+O). *The upper bound surrogate loss function* $L_{upper}(\hat{\mathbf{c}}, \mathbf{c})$ *(Elmachtoub & Grigas (2022)) is given by*

$$L_{upper}(\hat{\mathbf{c}}, \mathbf{c}) = - \min_{\mathbf{z} \in \mathbf{W}} \left\{ (2\hat{\mathbf{c}} - \mathbf{c})^\top \mathbf{z} \right\} + 2\hat{\mathbf{c}}^\top \mathbf{z}^*(\mathbf{c}) - \mathbf{c}^\top \mathbf{z}^*(\mathbf{c}),$$

*is convex with respect to* $\hat{\mathbf{c}}$*. Moreover, the surrogate gradient*

$$g_{upper} = 2\mathbf{z}^*(\mathbf{c}) - 2\mathbf{z}^\star,$$

*where* $\mathbf{z}^\star \in \arg\min_{\mathbf{z} \in \mathbf{W}}(2\hat{\mathbf{c}} - \mathbf{c})^\top \mathbf{z}$*, is a subgradient of* $L_{upper}(\hat{\mathbf{c}}, \mathbf{c})$ *at* $\hat{\mathbf{c}}$*.*

Based on the above lemmas, which establish the convexity and subgradient properties of the surrogate functions, we proceed with the convergence analysis of AProx. Using Lemma 2, we directly update the surrogate gradient in conjunction with the continuous gradient $\nabla f(x)$, which simplifies the proximal update formulation.

We provide the following theorem under appropriate assumptions, demonstrating the convergence of AProx for the three surrogate methods introduced.

**Theorem 1.** *Assume that the function* $f : \mathbb{R}^d \to \mathbb{R}$ *is convex and differentiable, and that the subgradient* $\tilde{R}$ *is a convex, potentially non-smooth function. For all iterations* $k$*, the gradients and subgradients are bounded, and there exists* $G_\infty > 0$ *such that* $\|\nabla f(\hat{\mathbf{c}}_k)\|_\infty \leq G_\infty$ *and* $\|\tilde{g}_k\|_\infty \leq G_\infty$*. Assume* $\beta_1, \beta_2 \in [0, 1)$*, and they satisfy* $\frac{\beta_1^2}{\sqrt{\beta_2}} < 1$*, with a learning rate* $\alpha > 0$ *and weight decay coefficient* $\lambda \geq 0$*. The cumulative regret* $\mathcal{R}(T)$ *satisfies:*

$$\mathcal{R}(T) = \sum_{t=1}^T (F(\hat{\mathbf{c}}_t) - F(\hat{\mathbf{c}}^*)) \leq \frac{D^2}{2\alpha(1 - \beta_1)} \sum_{i=1}^d \sqrt{\hat{v}_{T,i}} + \frac{\alpha G_\infty^2}{(1 - \beta_1)^2 (1 - \beta_2)} T$$

Theorem 1 proves that under the assumptions of convexity and bounded gradient, AProx ensures convergence with appropriate bounds on the cumulative regret values. Even in non-smooth generational gradients, the outlined conditions ensure that AProx remains stable throughout the iterations, thus effectively addressing the challenges inherent in the prediction+optimisation framework. The detailed proof of this theorem and supporting lemmas can be found in Appendix A.5 for further reference.

## 5 RELATED WORKS

**Predict+Optimize** The Predict+Optimize problem aims to solve a class of parametric optimization problems in an end-to-end manner, where a machine learning model predicts the optimization problem parameters. The main challenge of the problem is that the loss function during end-to-end training is non-differentiable concerning the predicted parameters, making it infeasible to obtain the loss and back-propagate the gradient.

The paths to solving this problem so far can be broadly categorized into two groups: One is the differentiable layer implementations developed to solve a specific optimization problem and embedding the differentiable layer into a framework for end-to-end optimization. Existing research has been conducted for stochastic optimization (Donti et al. (2017)), quadratic programming (Amos & Kolter (2017)), integer programming (Mandi & Guns (2020)), constrained optimization (Donti et al. (2021); Hu et al. (2023a)) and logic programming (Nandwani et al. (2022)) have been extensively studied. Another path to solving the non-trivial loss function is to construct a surrogate function to obtain the corresponding gradient, and the main paths so far are designing convex upper bounds (Elmachtoub & Grigas (2022)), dynamic programming (Stuckey et al. (2020)), decision trees (Elmachtoub et al. (2020)), black box approximation (Pogančić et al. (2020)), adding Gaussian perturbation (Berthet et al. (2020)), using the rank approach (Mandi et al. (2022)), contrast optimization approach (Mulamba et al. (2021)), linearization (Ferber et al. (2023)), etc..

Most current research focuses on gradient acquisition, but the gradient update process is underexplored. The gradient obtained by the surrogate function is not like the general end-to-end learning, where exact gradients can be easily obtained.

**Inexact Proximal Gradient Methods** The proximal gradient method has been investigated in a variety of optimization problems for the problem of inexact gradient and is regarded as one of the means to efficiently handle the inexact gradient. In recent years, Ajalloeian et al. (2020) extended this concept by developing an inexact online proximal-gradient method tailored for time-varying convex optimization problems. Bastianello & Dall'Anese (2021) introduced a distributed and inexact proximal gradient method specifically designed for online convex optimization. Moreover, Barré et al. (2023) provided a comprehensive analysis of first-order methods with inexact proximal operators. Yang & Li (2023) focused on using the Kurdyka-Łojasiewicz (KL) property to ensure convergence in nonconvex and nonsmooth optimization problems.

Their work demonstrated that the use of the inexact proximal gradient can keep the optimization process stable, inspiring our work to extend the proximal gradient approach.

**Optimizer** In the context of large-scale data for artificial intelligence, many variants of stochastic gradient descent (SGD) algorithms have been developed to improve the convergence performance, such as: vSGD (Schaul et al. (2013)), The Sum-of-Functions Optimizer (SFO) (Sohl-Dickstein et al. (2014)), and the well-known Adam optimizer by (Kingma (2014)).

With a variety of end-to-end learning tasks being proposed, optimizers are still being investigated in recent years to adapt to the characteristics of different learning tasks. Sun et al. (2020) explored gradient descent learning with "floats". Demidovich et al. (2023) provided a detailed guide through the diverse landscape of biased stochastic gradient descent (SGD) methods. Wang & Chen (2024) took a step further by analyzing the stability and generalization bounds in decentralized minibatch stochastic gradient descent.

As previously discussed, no optimizer has been developed to date for the characteristics of the P+O framework, and the resulting Exact gradient problem has no clear solution.

# 6 EXPERIMENTS

## 6.1 EXPERIMENT SETTINGS

We implement our codes primarily using Gurobi (Gurobi Optimization, LLC (2023)) and PyTorch (Paszke et al. (2019)), with additional help from PyEPO (Tang & Khalil (2022)). All experiments are conducted in a consistent computational environment featuring an Intel i7 CPU, 32GB of RAM, and an NVIDIA RTX 4070 Ti GPU.

**Baseline Optimizers** In our experiments, we compare AProx with several state-of-the-art optimizers, including AdaGrad (Duchi et al. (2011)), RMSProp (Tieleman & Hinton (2012)), AdaDelta (Zeiler (2012)), Adam (Kingma (2014)), and AdamW (Loshchilov & Hutter (2017)). These optimizers serve as baselines to evaluate the effectiveness of AProx in updating gradients and parameters during training across various benchmarks.

**Surrogate Functions** To thoroughly evaluate whether the AProx is valid across various surrogate gradient, we conducted experiments on both convex and non-convex surrogate functions within the

Predict+Optimize (P+O) framework. Specifically, we use five surrogate solutions to obtain inexact gradients on each benchmark, including three convex surrogate loss functions, IMLE (Niepert et al. (2021)), CMAP (Mulamba et al. (2021)), and SPO (Elmachtoub & Grigas (2022)), whose resulting gradients have been proven to be subgradients. In addition, we selected two non-convex approaches, DBB (Pogančić et al. (2020)) and NID (Sahoo et al.), to further validate the broader applicability and robustness of our optimizer. This diverse selection ensures that our experimental results encompass a wide range of gradient behaviors, from theoretically well-understood convex settings to more challenging non-convex scenarios.

**Parameter Settings** All experiments used consistent settings across benchmarks. The learning rate ranged from 1e-5 to 1e-3, and random seeds were fixed at 2024 for reproducibility. Training ran for up to 50 epochs, with a convergence threshold of 1e-2. These settings ensure that performance differences are due to the optimizers, not experimental variations.

## 6.2 BENCHMARKS DESCRIPTION

**Production Sales Problem (Sales)** The Sales problem is a variant of the 0-1 knapsack problem (Hu et al. (2023a)), focused on optimizing real estate investments. Investors select housing projects under budget constraints to maximize predicted profits. The decision variable $x_h$ represents whether to invest in project $h$. Given construction costs $c_h$, predicted sales prices $p_h$, and budget $B$, the objective is:

$$\max_{x_h} \sum_{h \in H} p_h x_h \quad \text{s.t.} \quad \sum_{h \in H} c_h x_h \leq B, \quad x_h \in \{0, 1\}$$

**Portfolio Problem (Portfolio)** In the Portfolio problem (Tang & Khalil (2022)), the goal is to allocate investments across assets to maximize expected returns while managing risk. The decision variable $x_i$ represents the proportion of asset investment $i$, with expected return $r_i$ and risk captured by the covariance matrix $C$. The problem is formulated as:

$$\max_{x_i} \sum_{i=1}^{n} r_i x_i \quad \text{s.t.} \quad \sum_{i=1}^{n} x_i = 1, \quad x^T C x \leq \gamma, \quad x_i \geq 0$$

**Shortest Path Problem (Path)** The Path problem (Tang & Khalil (2022)) aims to find the lowest-cost path from a source to a destination node in a network. The decision variable $x_{ij}$ represents the flow along arc $(i, j)$. The objective is to minimize the total traversal cost:

$$\min_{x_{ij}} \sum_{(i,j) \in A} c_{ij} x_{ij}, \quad \text{s.t.} \quad \sum_{(i,v) \in A} x_{iv} - \sum_{(v,j) \in A} x_{vj} = \begin{cases} -1 & \text{if } v = s \\ 1 & \text{if } v = t \\ 0 & \text{otherwise} \end{cases} \quad x_{ij} \geq 0, \forall (i,j) \in A$$

Here, $A$ represents the set of arcs, and $c_{ij}$ denotes the travel cost from node $i$ to node $j$.

## 6.3 RESULTS DISCUSSION

In this section, we present the comprehensive performance of AProx evaluated across different surrogate gradients on three benchmark problems: Sales, Portfolio, and Path. Specifically, we assess AProx's convergence performance, its optimal gap performance compared to baseline optimizers, and the findings from our ablation studies.

**Convergence Performance:** TTable 1 presents the average convergence performance of baseline optimizers applied to all five surrogate gradients (IMLE, CMAP, SPO, DBB, NID) across three benchmarks. The evaluation includes both the number of epochs required for convergence and the average time per epoch. Across all three benchmarks, AProx consistently demonstrates superior performance in terms of convergence speed, reflected by fewer epochs required on average.

For instance, in the Sales benchmark, AProx achieves the lowest average epochs (19.85 ± 21.29) compared to other optimizers, while maintaining competitive time per epoch. In the Portfolio benchmark, AProx not only shows fewer epochs (20.62 ± 21.36) but also reports a lower training time per epoch (151.98 ± 209.12), highlighting its efficiency and adaptability. For the Path benchmark,

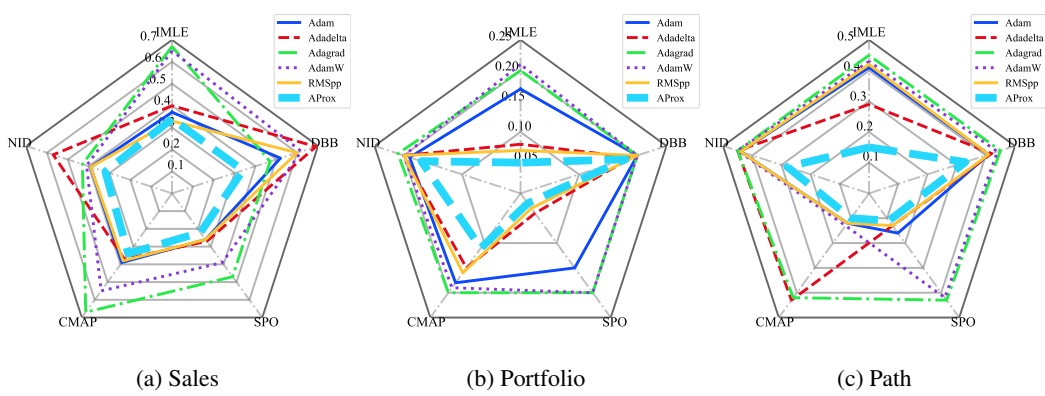

(a) Sales        (b) Portfolio        (c) Path

Figure 1: Optimal gap comparison of AProx against baseline optimizers (Adam, Adadelta, Adagrad, AdamW, RMSpp) across five surrogate gradients (IMLE, CMAP, SPO, DBB, NID) for three benchmark problems: (a) Sales, (b) Portfolio, and (c) Path.

AProx significantly reduces the convergence epochs to just ($6.84 \pm 2.46$), setting it apart from other optimizers, which require substantially more epochs.

These results indicate that AProx balances fewer training epochs with reasonable computational costs, making it suitable for scenarios needing fast convergence and strong optimal gap performance.

**Baseline Comparisons:** After conducting convergence experiments, we compare the performance gap of AProx with other optimizers when using different regret functions, as shown in Figure 1. The metrics in these radar plots represent the ratio of the distance between the optimal solution based on the predicted and the true parameters, divided by the true optimal solution. The closer an optimizer's performance line is to the center, the better it performs across all metrics.

In the Sales benchmark (subfigure (a)), AProx remains consistently close to the center across multiple metrics (IMLE, NID, CMAP, SPO, DBB), highlighting its superior performance compared to baselines like Adam and Adadelta, which display greater variability. AProx's balanced and compact shape suggests an overall stronger performance.

Subfigure (b) of Figure 1 presents the portfolio problem, where AProx maintains a favorable position with lines consistently near the center, particularly excelling in metrics such as SPO

|  | Optimizer | Convergence Performance | |
|---|---|---|---|
|  |  | Epochs | Time per Epoch |
| Sales | Adam | $28.40 \pm 19.71$ | $\mathbf{25.65 \pm 23.91}$ |
|  | Adadelta | $30.80 \pm 19.73$ | $35.76 \pm 59.85$ |
|  | Adagrad | $24.80 \pm 22.25$ | $27.06 \pm 35.28$ |
|  | AdamW | $37.80 \pm 15.74$ | $44.58 \pm 61.20$ |
|  | RMSpp | $23.80 \pm 23.13$ | $38.72 \pm 52.65$ |
|  | **AProx** | $\mathbf{19.85 \pm 21.29}$ | $33.87 \pm 41.43$ |
| Portfolio | Adam | $37.80 \pm 17.88$ | $227.29 \pm 256.42$ |
|  | Adadelta | $26.60 \pm 21.59$ | $190.48 \pm 237.08$ |
|  | Adagrad | $23.80 \pm 23.08$ | $191.12 \pm 233.44$ |
|  | AdamW | $33.20 \pm 21.78$ | $228.82 \pm 258.10$ |
|  | RMSpp | $31.40 \pm 18.34$ | $230.14 \pm 258.47$ |
|  | **AProx** | $\mathbf{20.62 \pm 21.36}$ | $\mathbf{151.98 \pm 209.12}$ |
| Path | Adam | $23.20 \pm 17.56$ | $27.28 \pm 27.71$ |
|  | Adadelta | $22.60 \pm 24.10$ | $39.53 \pm 48.34$ |
|  | Adagrad | $22.60 \pm 24.10$ | $39.50 \pm 48.33$ |
|  | AdamW | $22.60 \pm 17.36$ | $27.48 \pm 27.95$ |
|  | RMSpp | $14.60 \pm 10.36$ | $22.41 \pm 23.31$ |
|  | **AProx** | $\mathbf{6.84 \pm 2.46}$ | $\mathbf{20.70 \pm 23.62}$ |

Table 1: Average convergence performance of baseline optimizers across all five surrogate gradients (IMLE, CMAP, SPO, DBB, NID) on three benchmarks.

and CMAP. Compared to Adam, Adadelta, and Adagrad, whose performance shows higher deviation, AProx delivers a more robust and balanced outcome across all dimensions.

In subfigure (c), which represents the shortest path benchmark, AProx once again stays close to the center, indicating better overall performance across all surrogate metrics. In contrast, Adam and Adadelta show lines farther from the center, particularly in SPO and CMAP, suggesting poorer results relative to AProx.

|  | Optimizer | Convex | | | Non-convex | |
|---|---|---|---|---|---|---|
|  |  | IMLE | CMAP | SPO | DBB | NID |
| Sales | **AProx** | **0.34** | **0.32** | **0.22** | **0.34** | **0.34** |
|  | AProx_NoProx | **0.62** (82.35%) | **0.64** (100.00%) | 0.36 (63.64%) | **0.47** (38.24%) | **0.62** (82.35%) |
|  | AProx_NoAdaptive | 0.51 (50.00%) | 0.36 (12.50%) | 0.28 (27.27%) | 0.42 (23.53%) | 0.36 (5.88%) |
|  | AProx_NoMomentum | 0.49 (44.12%) | **0.62** (93.75%) | **0.38** (72.73%) | 0.43 (26.47%) | 0.61 (79.41%) |
|  | AProx_NoWeightDecay | 0.53 (55.88%) | 0.34 (6.25%) | 0.33 (50.00%) | 0.39 (14.71%) | 0.61 (79.41%) |
| Portfolio | **AProx** | **0.05** | **0.11** | **0.02** | **0.17** | **0.18** |
|  | AProx_NoProx | **0.18** (260.00%) | **0.29** (163.64%) | **0.18** (800.00%) | **0.19** (11.76%) | 0.18 (0.00%) |
|  | AProx_NoAdaptive | 0.16 (220.00%) | 0.13 (18.18%) | 0.17 (750.00%) | 0.17 (0.00%) | 0.18 (0.00%) |
|  | AProx_NoMomentum | **0.18** (260.00%) | 0.10 (-9.09%) | 0.12 (500.00%) | 0.18 (5.88%) | **0.19** (5.56%) |
|  | AProx_NoWeightDecay | 0.15 (200.00%) | 0.13 (18.18%) | 0.03 (50.00%) | 0.17 (0.00%) | 0.18 (0.00%) |
| Path | **AProx** | **0.15** | **0.09** | **0.11** | **0.33** | **0.28** |
|  | AProx_NoProx | **0.40** (166.67%) | 0.14 (55.56%) | **0.13** (18.18%) | 0.35 (6.06%) | **0.40** (42.86%) |
|  | AProx_NoAdaptive | 0.16 (6.67%) | **0.31** (244.44%) | 0.12 (9.09%) | **0.36** (9.09%) | 0.34 (21.43%) |
|  | AProx_NoMomentum | 0.31 (106.67%) | 0.12 (33.33%) | 0.11 (0.00%) | 0.35 (6.06%) | 0.32 (14.29%) |
|  | AProx_NoWeightDecay | 0.28 (86.67%) | 0.10 (11.11%) | 0.11 (0.00%) | 0.33 (0.00%) | 0.29 (3.57%) |

Table 2: Ablation study results showing the optimal gaps for AProx and its variants across different surrogate gradients on three benchmarks. The percentages indicate the increase of the optimal gap compared to AProx.

Therefore, AProx demonstrates a consistent advantage in minimizing the optimal gap across all benchmarks and surrogate functions, reinforcing its effectiveness compared to traditional optimizers. More detailed results are available in Table 4 in Appendix C.

**Ablation Study:** Finally, we would like to go a step further and verify how much the proposed modules in AProx contribute to performance improvement. Table 2 presents the ablation results of AProx compared to its variations, AProx_NoProx, AProx_NoAdaptive, AProx_NoMomentum, and AProx_NoWeightDecay, on different surrogate gradients across three benchmarks: Sales, Portfolio, and Path.

For the Sales benchmark, AProx achieves the lowest optimal gaps across all surrogate functions, such as 0.34 for IMLE and 0.22 for SPO. Removing the proximal component (AProx_NoProx) increases the optimal gap by 82.35% for IMLE, demonstrating its effectiveness. A similar trend holds across other ablated versions, with AProx always performing better.

In the Portfolio benchmark, removing the proximal component leads to significant increases in the optimal gap (260% for IMLE, 800% for SPO), showing its crucial role in reducing sub-optimality. Both the proximal gradient and momentum significantly contribute to AProx's performance in this benchmark. It is important to note here that this large scaling up is due to the small optimal gap of AProx under this problem.

For the Path benchmark, AProx achieves the lowest optimal gap across most metrics, particularly for IMLE (0.15) and CMAP (0.09). Removing adaptive or proximal components results in larger gaps, such as a 166.67% increase for IMLE and 244.44% for CMAP.

Overall, these results indicate that the remove of components increases the gap, highlighting the importance of each part for robust optimization.

# 7 CONCLUSION

This work introduces the Adaptive Proximal Gradient Optimizer (AProx) to address gradient inexactness in the Predict+Optimize (P+O) framework. AProx effectively handles inaccuracies from surrogate gradients and achieves convergence speeds similar to smooth optimization methods through composite function and proximal gradient techniques. We further enhance it with adaptive learning rates, momentum, weight decay, and parameter averaging, improving performance beyond traditional gradient descent. Experiments on combinatorial benchmarks show that AProx accelerates convergence and outperforms methods that overlook gradient inexactness. However, the issue of inexact gradients remains under-explored, presenting opportunities for future research to strengthen theoretical understanding and develop more robust optimization techniques for P+O.

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

## A  THEORETICAL RESULTS AND PROOFS

### A.1  PROOF OF LEMMA 1

**Lemma 1** (Implict Proximal Gradient Descent Update rule). *Let $\tilde{R} : \mathbb{R}^n \to \mathbb{R}$ be a convex, non-smooth function, and let $f : \mathbb{R}^n \to \mathbb{R}$ be a differentiable convex function with an L-Lipschitz continuous gradient. The proximal gradient descent update formula $\hat{\boldsymbol{c}}_{t+1} = \mathrm{prox}_{\eta\tilde{R}}\left(\hat{\boldsymbol{c}}_t - \eta\nabla f(\hat{\boldsymbol{c}}_t)\right)$ can be equivalently written in the form:*

$$\hat{\boldsymbol{c}}_{t+1} = \hat{\boldsymbol{c}}_t - \eta\nabla f(\hat{\boldsymbol{c}}_t) - \eta\tilde{g}_t,$$

*where $\tilde{g}_t \in \partial\tilde{R}(\hat{\boldsymbol{c}}_{t+1})$ represents a subgradient of $\tilde{R}$ at $\hat{\boldsymbol{c}}_{t+1}$.*

*Proof* By the definition of the proximal operator, we have:

$$\hat{\boldsymbol{c}}_{t+1} = \mathrm{prox}_{\eta\tilde{R}}\left(\hat{\boldsymbol{v}}\right),$$

where

$$\hat{\boldsymbol{v}} = \hat{\boldsymbol{c}}_t - \eta\nabla f(\hat{\boldsymbol{c}}_t).$$

This means that $\hat{\boldsymbol{c}}_{t+1}$ is the minimizer of the following optimization problem:

$$\hat{\boldsymbol{c}}_{t+1} = \arg\min_{\hat{\boldsymbol{c}}} \left\{ \tilde{R}(\hat{\boldsymbol{c}}) + \frac{1}{2\eta} \|\hat{\boldsymbol{c}} - \hat{\boldsymbol{v}}\|^2 \right\}.$$

From the first-order optimality condition for convex functions, we have:

$$\mathbf{0} \in \partial \tilde{R}(\hat{\boldsymbol{c}}_{t+1}) + \frac{1}{\eta} \left( \hat{\boldsymbol{c}}_{t+1} - \hat{\boldsymbol{v}} \right),$$

where $\partial \tilde{R}(\hat{\boldsymbol{c}}_{t+1})$ denotes the subdifferential of $\tilde{R}$ at $\hat{\boldsymbol{c}}_{t+1}$.

Substituting $\hat{\boldsymbol{v}} = \hat{\boldsymbol{c}}_t - \eta \nabla f(\hat{\boldsymbol{c}}_t)$, we get:

$$\mathbf{0} \in \partial \tilde{R}(\hat{\boldsymbol{c}}_{t+1}) + \frac{1}{\eta} \left( \hat{\boldsymbol{c}}_{t+1} - (\hat{\boldsymbol{c}}_t - \eta \nabla f(\hat{\boldsymbol{c}}_t)) \right).$$

Simplifying the expression inside the parentheses:

$$\hat{\boldsymbol{c}}_{t+1} - \hat{\boldsymbol{c}}_t + \eta \nabla f(\hat{\boldsymbol{c}}_t) = \hat{\boldsymbol{c}}_{t+1} - \hat{\boldsymbol{c}}_t + \eta \nabla f(\hat{\boldsymbol{c}}_t).$$

Therefore, the optimality condition becomes:

$$\mathbf{0} \in \partial \tilde{R}(\hat{\boldsymbol{c}}_{t+1}) + \frac{1}{\eta} \left( \hat{\boldsymbol{c}}_{t+1} - \hat{\boldsymbol{c}}_t + \eta \nabla f(\hat{\boldsymbol{c}}_t) \right).$$

Multiplying both sides by $\eta$:

$$\mathbf{0} \in \eta \, \partial \tilde{R}(\hat{\boldsymbol{c}}_{t+1}) + \left( \hat{\boldsymbol{c}}_{t+1} - \hat{\boldsymbol{c}}_t + \eta \nabla f(\hat{\boldsymbol{c}}_t) \right).$$

Rewriting the equation:

$$\hat{\boldsymbol{c}}_{t+1} = \hat{\boldsymbol{c}}_t - \eta \nabla f(\hat{\boldsymbol{c}}_t) - \eta \tilde{g}_t,$$

where $\tilde{g}_t \in \partial \tilde{R}(\hat{\boldsymbol{c}}_{t+1})$.

This demonstrates that the proximal gradient descent update can be expressed as a standard gradient descent step on $f$ followed by a subgradient step on $\tilde{R}$.

$\square$

## A.2 PROOF OF LEMMA 2

**Lemma 2** (Surrogate Gradient of Perturbed Methods IN P+O). *The perturbed surrogate loss function $L_{pert}(\mathbf{c}, \hat{\mathbf{c}})$ (Niepert et al. (2021)) is given by*

$$L_{pert}(\mathbf{c}, \hat{\mathbf{c}}) = \mathbb{E}_{\hat{\mathbf{z}} \sim q(\mathbf{z}; \hat{\mathbf{c}})} \left[ A(\mathbf{c}) - \langle \hat{\mathbf{z}}, \mathbf{c} \rangle \right],$$

*is convex with respect to $\mathbf{c}$. Moreover, the surrogate gradient*

$$g_{pert} = \boldsymbol{\mu}(\mathbf{c}) - \boldsymbol{\mu}(\hat{\mathbf{c}}),$$

*where $\boldsymbol{\mu}(\mathbf{c}) = \nabla_{\mathbf{c}} A(\mathbf{c})$, is a subgradient of $L_{pert}(\mathbf{c}, \hat{\mathbf{c}})$ at $\mathbf{c}$.*

*Proof* The function $A(\mathbf{c})$ is the log-partition function of an exponential family distribution, which is known to be convex in $\mathbf{c}$. The second term, $\langle \hat{\mathbf{z}}, \mathbf{c} \rangle$, is linear in $\mathbf{c}$. Since the expectation of a convex function remains convex, $L_{\text{pert}}(\mathbf{c}, \hat{\mathbf{c}})$ is convex in $\mathbf{c}$.

The gradient of $L_{\text{pert}}$ with respect to $\mathbf{c}$ can be computed as:

$$\nabla_{\mathbf{c}} L_{\text{pert}}(\mathbf{c}, \hat{\mathbf{c}}) = \nabla_{\mathbf{c}} A(\mathbf{c}) - \mathbb{E}_{\hat{\mathbf{z}} \sim q(\mathbf{z}; \hat{\mathbf{c}})}[\hat{\mathbf{z}}] = \boldsymbol{\mu}(\mathbf{c}) - \boldsymbol{\mu}(\hat{\mathbf{c}}).$$

For any $\mathbf{c}' \in \mathbb{R}^n$, by the convexity of $A(\mathbf{c})$:

$$L_{\text{pert}}(\mathbf{c}', \hat{\mathbf{c}}) \geq L_{\text{pert}}(\mathbf{c}, \hat{\mathbf{c}}) + (\boldsymbol{\mu}(\mathbf{c}) - \boldsymbol{\mu}(\hat{\mathbf{c}}))^{\top}(\mathbf{c}' - \mathbf{c}),$$

which verifies that $g_{\text{pert}} = \boldsymbol{\mu}(\mathbf{c}) - \boldsymbol{\mu}(\hat{\mathbf{c}})$ is a subgradient of $L_{\text{pert}}(\mathbf{c}, \hat{\mathbf{c}})$ at $\mathbf{c}$.

$\square$

### A.3 PROOF OF LEMMA 3

**Lemma 3** (Surrogate Gradient of Contrastive Methods IN P+O). *The CMAP surrogate loss function* $L_{contrast}(\hat{\mathbf{c}}, \mathbf{c})$ *(Mulamba et al. (2021)) is given by*

$$L_{contrast}(\hat{\mathbf{c}}, \mathbf{c}) = \frac{1}{|\Gamma| - 1} \sum_{\mathbf{z} \in \Gamma \setminus \{\mathbf{z}^*(\mathbf{c})\}} \left( \hat{\mathbf{c}}^{\top} \mathbf{z}^*(\mathbf{c}) - \hat{\mathbf{c}}^{\top} \mathbf{z} \right),$$

*is convex with respect to* $\hat{\mathbf{c}}$. *Moreover, the surrogate gradient*

$$g_{contrast} = \frac{1}{|\Gamma| - 1} \sum_{\mathbf{z} \in \Gamma \setminus \{\mathbf{z}^*(\mathbf{c})\}} (\mathbf{z}^*(\mathbf{c}) - \mathbf{z})$$

*is a subgradient of* $L_{contrast}(\hat{\mathbf{c}}, \mathbf{c})$ *at* $\hat{\mathbf{c}}$.

*Proof* The function $L_{\text{contrast}}(\hat{\mathbf{c}}, \mathbf{c})$ is an average of linear terms of the form $\hat{\mathbf{c}}^{\top}(\mathbf{z}^*(\mathbf{c}) - \mathbf{z})$, which are all linear in $\hat{\mathbf{c}}$. Since linear functions are both convex and concave, $L_{\text{contrast}}$ is convex in $\hat{\mathbf{c}}$.

The surrogate gradient can be computed as:

$$g_{\text{contrast}} = \frac{1}{|\Gamma| - 1} \sum_{\mathbf{z} \in \Gamma \setminus \{\mathbf{z}^*(\mathbf{c})\}} (\mathbf{z}^*(\mathbf{c}) - \mathbf{z}).$$

For any $\hat{\mathbf{c}}' \in \mathbb{R}^n$:

$$L_{\text{contrast}}(\hat{\mathbf{c}}', \mathbf{c}) - L_{\text{contrast}}(\hat{\mathbf{c}}, \mathbf{c}) = g_{\text{contrast}}^{\top}(\hat{\mathbf{c}}' - \hat{\mathbf{c}}),$$

which verifies that $g_{\text{contrast}}$ is a subgradient of $L_{\text{contrast}}(\hat{\mathbf{c}}, \mathbf{c})$ at $\hat{\mathbf{c}}$.

$\square$

### A.4 PROOF OF LEMMA 4

**Lemma 4** (Surrogate Gradient of Upper Bound Methods IN P+O). *The upper bound surrogate loss function* $L_{upper}(\hat{\mathbf{c}}, \mathbf{c})$ *(Elmachtoub & Grigas (2022)) is given by*

$$L_{upper}(\hat{\mathbf{c}}, \mathbf{c}) = -\min_{\mathbf{z} \in \mathbf{W}} \left\{ (2\hat{\mathbf{c}} - \mathbf{c})^{\top} \mathbf{z} \right\} + 2\hat{\mathbf{c}}^{\top} \mathbf{z}^*(\mathbf{c}) - \mathbf{c}^{\top} \mathbf{z}^*(\mathbf{c}),$$

*is convex with respect to* $\hat{\mathbf{c}}$. *Moreover, the surrogate gradient*

$$g_{upper} = 2\mathbf{z}^*(\mathbf{c}) - 2\mathbf{z}^{\star},$$

*where* $\mathbf{z}^{\star} \in \arg\min_{\mathbf{z} \in \mathbf{W}} (2\hat{\mathbf{c}} - \mathbf{c})^{\top} \mathbf{z}$, *is a subgradient of* $L_{upper}(\hat{\mathbf{c}}, \mathbf{c})$ *at* $\hat{\mathbf{c}}$.

*Proof* To prove convexity, we first rewrite the given function $L_{\text{upper}}(\hat{\mathbf{c}}, \mathbf{c})$. Notice that the term involving the minimum can be expressed as a maximization:

$$-\min_{\mathbf{z} \in \mathbf{W}} \left\{ (2\hat{\mathbf{c}} - \mathbf{c})^\top \mathbf{z} \right\} = \max_{\mathbf{z} \in \mathbf{W}} \left\{ -(2\hat{\mathbf{c}} - \mathbf{c})^\top \mathbf{z} \right\}.$$

Thus, the loss function can be reformulated as:

$$L_{\text{upper}}(\hat{\mathbf{c}}, \mathbf{c}) = \max_{\mathbf{z} \in \mathbf{W}} \left\{ -(2\hat{\mathbf{c}} - \mathbf{c})^\top \mathbf{z} \right\} + 2\hat{\mathbf{c}}^\top \mathbf{z}^*(\mathbf{c}) - \mathbf{c}^\top \mathbf{z}^*(\mathbf{c}).$$

The first term, $\max_{\mathbf{z} \in \mathbf{W}} \left\{ -(2\hat{\mathbf{c}} - \mathbf{c})^\top \mathbf{z} \right\}$, represents the pointwise maximum of affine functions of $\hat{\mathbf{c}}$, which is a convex operation. The second term, $2\hat{\mathbf{c}}^\top \mathbf{z}^*(\mathbf{c})$, is affine in $\hat{\mathbf{c}}$, hence convex. The third term, $-\mathbf{c}^\top \mathbf{z}^*(\mathbf{c})$, is constant with respect to $\hat{\mathbf{c}}$ and does not affect the convexity. Therefore, $L_{\text{upper}}(\hat{\mathbf{c}}, \mathbf{c})$ is convex with respect to $\hat{\mathbf{c}}$.

For the subgradient, we consider the optimal solution $\mathbf{z}^\star$, which minimizes $(2\hat{\mathbf{c}} - \mathbf{c})^\top \mathbf{z}$ over $\mathbf{z} \in \mathbf{W}$. The surrogate gradient can be computed by taking the gradient of the affine components:

$$g_{\text{upper}} = \nabla_{\hat{\mathbf{c}}} \left( -(2\hat{\mathbf{c}} - \mathbf{c})^\top \mathbf{z}^\star + 2\hat{\mathbf{c}}^\top \mathbf{z}^*(\mathbf{c}) \right) = -2\mathbf{z}^\star + 2\mathbf{z}^*(\mathbf{c}) = 2\mathbf{z}^*(\mathbf{c}) - 2\mathbf{z}^\star.$$

To verify that $g_{\text{upper}}$ is a subgradient, consider any $\hat{\mathbf{c}}' \in \mathbb{R}^n$:

$$L_{\text{upper}}(\hat{\mathbf{c}}', \mathbf{c}) \geq L_{\text{upper}}(\hat{\mathbf{c}}, \mathbf{c}) + g_{\text{upper}}^\top (\hat{\mathbf{c}}' - \hat{\mathbf{c}}).$$

Since $\mathbf{z}^\star$ is an optimal solution, this inequality holds, confirming that $g_{\text{upper}}$ is a subgradient of $L_{\text{upper}}(\hat{\mathbf{c}}, \mathbf{c})$ at $\hat{\mathbf{c}}$.

$\square$

## A.5 PROOF OF THEOREM 1

**Lemma 5** (Convexity). *For any* $\boldsymbol{x}, \boldsymbol{y} \in \mathbb{R}^d$:

$$f(\boldsymbol{y}) \geq f(\boldsymbol{x}) + \nabla f(\boldsymbol{x})^T (\boldsymbol{y} - \boldsymbol{x})$$

*Proof* This is a fundamental property of convex functions. $\square$

**Lemma 6** (Subgradient Inequality for $\tilde{R}$). *For any* $\boldsymbol{x}, \boldsymbol{y} \in \mathbb{R}^d$, $\tilde{\boldsymbol{g}}_x \in \partial \tilde{R}(\boldsymbol{x})$, *where* $\tilde{\boldsymbol{g}}_x = \tilde{\boldsymbol{g}}'_x + \boldsymbol{\delta}_x$, *and* $\|\boldsymbol{\delta}_x\|_2 \leq \delta$:

$$\tilde{R}(\boldsymbol{y}) \geq \tilde{R}(\boldsymbol{x}) + \boldsymbol{g}_x^T (\boldsymbol{y} - \boldsymbol{x}) - \boldsymbol{\delta}_x^T (\boldsymbol{y} - \boldsymbol{x})$$

*Proof* Since $\tilde{R}$ is convex, for any $\boldsymbol{g}_x \in \partial \tilde{R}(\boldsymbol{x})$:

$$\tilde{R}(\boldsymbol{y}) \geq \tilde{R}(\boldsymbol{x}) + \boldsymbol{g}_x^T (\boldsymbol{y} - \boldsymbol{x})$$

Given $\tilde{\boldsymbol{g}}_x = \tilde{\boldsymbol{g}}'_x + \boldsymbol{\delta}_x$:

$$\tilde{R}(\boldsymbol{y}) \geq \tilde{R}(\boldsymbol{x}) + \left( \tilde{\boldsymbol{g}}'_x - \boldsymbol{\delta}_x \right)^T (\boldsymbol{y} - \boldsymbol{x}) = \tilde{R}(\boldsymbol{x}) + \tilde{\boldsymbol{g}}'^T_x (\boldsymbol{y} - \boldsymbol{x}) - \boldsymbol{\delta}_x^T (\boldsymbol{y} - \boldsymbol{x})$$

$\square$

**Lemma 7** (Bound on the Sum of Scaled Gradients). *For each coordinate* $i$, *if* $\|\nabla f(\hat{\boldsymbol{c}}_k)\|_\infty \leq G_\infty$, $\|\tilde{\boldsymbol{g}}_k\|_\infty \leq G_\infty$, *and* $\hat{v}_k \geq (1 - \beta_2) d_{k,i}^2$:

$$\sum_{t=1}^{T} \frac{g_{t,i}^2}{\sqrt{\hat{v}_{t,i}}} \leq \frac{G_\infty^2}{(1 - \beta_2)\sqrt{1 - \beta_2}} T$$

*Proof*

Since $\hat{v}_{t,i} = \max(\hat{v}_{t-1,i}, v_{t,i}) \geq v_{t,i}$, and:

$$v_{t,i} = \beta_2 v_{t-1,i} + (1 - \beta_2)g_{t,i}^2 \geq (1 - \beta_2)g_{t,i}^2$$

Therefore:

$$\hat{v}_{t,i} \geq (1 - \beta_2)g_{t,i}^2$$

Then, Substituting the lower bound of $\hat{v}_{t,i}$:

$$\frac{g_{t,i}^2}{\sqrt{\hat{v}_{t,i}}} \leq \frac{g_{t,i}^2}{\sqrt{(1 - \beta_2)}|g_{t,i}|} = \frac{|g_{t,i}|}{\sqrt{1 - \beta_2}}$$

Since $|g_{t,i}| \leq 2G_\infty$, we have:

$$\sum_{t=1}^{T} \frac{g_{t,i}^2}{\sqrt{\hat{v}_{t,i}}} \leq \frac{2G_\infty}{\sqrt{1 - \beta_2}}T$$

Furthermore, since $2G_\infty \leq G_\infty^2/\sqrt{1 - \beta_2}$ for $G_\infty \geq 1$, we can write:

$$\sum_{t=1}^{T} \frac{g_{t,i}^2}{\sqrt{\hat{v}_{t,i}}} \leq \frac{G_\infty^2}{(1 - \beta_2)\sqrt{1 - \beta_2}}T$$

$\square$

**Lemma 8** (Bound on the Sum of Adaptive Learning Rates)**.** *For each coordinate $i$, given $\hat{m}_k = \frac{m_k}{1 - \beta_1^k}$:*

$$\sum_{t=1}^{T} \frac{(\hat{m}_{t,i})^2}{\sqrt{\hat{v}_{t,i}}} \leq \frac{G_\infty^2}{(1 - \beta_1)^2(1 - \beta_2)}T$$

*Proof*

From the definition in Algorithm 1:

$$\hat{m}_{t,i} = \frac{m_{t,i}}{1 - \beta_1^t}$$

Since $m_{t,i} = \beta_1 m_{t-1,i} + (1 - \beta_1)g_{t,i}$, unrolling:

$$m_{t,i} = (1 - \beta_1)\sum_{k=1}^{t} \beta_1^{t-k}g_{k,i}$$

Then:

$$|m_{t,i}| \leq (1 - \beta_1)\sum_{k=1}^{t} \beta_1^{t-k}|g_{k,i}| \leq (1 - \beta_1)\frac{(1 - \beta_1^t)|g_{k,i}|}{1 - \beta_1} \leq (1 - \beta_1)\frac{|g_{k,i}|}{1 - \beta_1}$$

Therefore:

$$|\hat{m}_{t,i}| \leq \frac{|g_{k,i}|}{1 - \beta_1}$$

From Lemma 7, we have:

$$\hat{v}_{t,i} \geq (1 - \beta_2)g_{t,i}^2$$

So:

$$\sqrt{\hat{v}_{t,i}} \geq \sqrt{1 - \beta_2}|g_{t,i}|$$

After, bounding the ratio:

$$\frac{(\hat{m}_{t,i})^2}{\sqrt{\hat{v}_{t,i}}} \leq \frac{\left(\frac{|g_{k,i}|}{1-\beta_1}\right)^2}{\sqrt{1 - \beta_2}|g_{t,i}|} = \frac{|g_{t,i}|}{(1 - \beta_1)^2\sqrt{1 - \beta_2}}$$

Summing Over $t$:

$$\sum_{t=1}^{T} \frac{(\hat{m}_{t,i})^2}{\sqrt{\hat{v}_{t,i}}} \leq \frac{1}{(1 - \beta_1)^2\sqrt{1 - \beta_2}} \sum_{t=1}^{T} |g_{t,i}| \leq \frac{2G_\infty T}{(1 - \beta_1)^2\sqrt{1 - \beta_2}}$$

Since $|g_{t,i}| \leq 2G_\infty$.

Recognizing that $\sqrt{1 - \beta_2} \leq 1$:

$$\sum_{t=1}^{T} \frac{(\hat{m}_{t,i})^2}{\sqrt{\hat{v}_{t,i}}} \leq \frac{2G_\infty T}{(1 - \beta_1)^2(1 - \beta_2)}$$

For the purposes of an upper bound, we can write:

$$\sum_{t=1}^{T} \frac{(\hat{m}_{t,i})^2}{\sqrt{\hat{v}_{t,i}}} \leq \frac{G_\infty^2}{(1 - \beta_1)^2(1 - \beta_2)}T$$

$\square$

**Theorem 1.** *Assume that the function $f : \mathbb{R}^d \rightarrow \mathbb{R}$ is convex and differentiable, and that the subgradient $\tilde{R}$ is a convex, potentially non-smooth function. For all iterations $k$, the gradients and subgradients are bounded, and there exists $G_\infty > 0$ such that $\|\nabla f(\hat{c}_k)\|_\infty \leq G_\infty$ and $\|\tilde{g}_k\|_\infty \leq G_\infty$. Assume $\beta_1, \beta_2 \in [0, 1)$, and they satisfy $\frac{\beta_1^2}{\sqrt{\beta_2}} < 1$, with a learning rate $\alpha > 0$ and weight decay coefficient $\lambda \geq 0$. The cumulative regret $\mathcal{R}(T)$ satisfies:*

$$\mathcal{R}(T) = \sum_{t=1}^{T} (F(\hat{c}_t) - F(\hat{c}^*)) \leq \frac{D^2}{2\alpha(1 - \beta_1)} \sum_{i=1}^{d} \sqrt{\hat{v}_{T,i}} + \frac{\alpha G_\infty^2}{(1 - \beta_1)^2(1 - \beta_2)}T$$

*Proof*

From Lemma 5 and Lemma 6, and considering the inexactness of subgradients, we have:

$$F(\hat{c}_t) - F(\hat{c}^*) \leq (\boldsymbol{g}_t - \boldsymbol{\delta}_t)^T (\hat{c}_t - \hat{c}^*)$$

Assuming $\|\boldsymbol{\delta}_t\|_2 \leq \delta$, we can write:

$$F(\hat{c}_t) - F(\hat{c}^*) \leq \boldsymbol{g}_t^T (\hat{c}_t - \hat{c}^*) + \delta\|\hat{c}_t - \hat{c}^*\|_2$$

Since $\|\hat{c}_t - \hat{c}^*\|_2 \leq D$, the error term due to inexactness is bounded.

From the update rule, we have:

$$\hat{\boldsymbol{c}}_{t+1} = \phi \left( \hat{\boldsymbol{c}}_t - \boldsymbol{\eta}_t \odot \hat{\boldsymbol{m}}_t \right),$$

where $\phi = \frac{1}{1+\alpha\lambda}$.

Compute the squared distance:

$$\|\hat{\boldsymbol{c}}_{t+1} - \hat{\boldsymbol{c}}^*\|_2^2 = \phi^2 \|\hat{\boldsymbol{c}}_t - \hat{\boldsymbol{c}}^* - \boldsymbol{\eta}_t \odot \hat{\boldsymbol{m}}_t\|_2^2$$

Expanding:

$$\|\hat{\boldsymbol{c}}_{t+1} - \hat{\boldsymbol{c}}^*\|_2^2 = \phi^2 \left( \|\hat{\boldsymbol{c}}_t - \hat{\boldsymbol{c}}^*\|_2^2 - 2(\hat{\boldsymbol{c}}_t - \hat{\boldsymbol{c}}^*)^T (\boldsymbol{\eta}_t \odot \hat{\boldsymbol{m}}_t) + \|\boldsymbol{\eta}_t \odot \hat{\boldsymbol{m}}_t\|_2^2 \right)$$

Then, subtract $\|\hat{\boldsymbol{c}}_t - \hat{\boldsymbol{c}}^*\|_2^2$:

$$\|\hat{\boldsymbol{c}}_{t+1} - \hat{\boldsymbol{c}}^*\|_2^2 - \|\hat{\boldsymbol{c}}_t - \hat{\boldsymbol{c}}^*\|_2^2 = (\phi^2 - 1)\|\hat{\boldsymbol{c}}_t - \hat{\boldsymbol{c}}^*\|_2^2 - 2\phi^2(\hat{\boldsymbol{c}}_t - \hat{\boldsymbol{c}}^*)^T (\boldsymbol{\eta}_t \odot \hat{\boldsymbol{m}}_t) + \phi^2 \|\boldsymbol{\eta}_t \odot \hat{\boldsymbol{m}}_t\|_2^2$$

From above, we have:

$$F(\hat{\boldsymbol{c}}_t) - F(\hat{\boldsymbol{c}}^*) \leq \boldsymbol{g}_t^T (\hat{\boldsymbol{c}}_t - \hat{\boldsymbol{c}}^*) + \delta D$$

We can relate $(\hat{\boldsymbol{c}}_t - \hat{\boldsymbol{c}}^*)^T (\boldsymbol{\eta}_t \odot \hat{\boldsymbol{m}}_t)$ to $F(\hat{\boldsymbol{c}}_t) - F(\hat{\boldsymbol{c}}^*)$. Assuming $\boldsymbol{\eta}_t$ and $\hat{\boldsymbol{m}}_t$ are aligned with $\boldsymbol{g}_t$, we have:

$$(\hat{\boldsymbol{c}}_t - \hat{\boldsymbol{c}}^*)^T (\eta_t \circ \hat{m}_t) = \sum_{i=1}^{d} (\hat{c}_{t,i} - \hat{c}_i^*)(\eta_{t,i} \hat{m}_{t,i}) = \sum_{i=1}^{d} (\hat{c}_{t,i} - \hat{c}_i^*) \left( \frac{\alpha \hat{m}_{t,i}}{\sqrt{\hat{v}_{t,i}} + \epsilon} \right) = \alpha \sum_{i=1}^{d} \frac{(\hat{c}_{t,i} - \hat{c}_i^*)\hat{m}_{t,i}}{\sqrt{\hat{v}_{t,i}} + \epsilon}$$

To proceed, we can use Cauchy-Schwarz inequality:

$$(\hat{\boldsymbol{c}}_t - \hat{\boldsymbol{c}}^*)^T (\boldsymbol{\eta}_t \odot \hat{\boldsymbol{m}}_t) \leq \|\hat{\boldsymbol{c}}_t - \hat{\boldsymbol{c}}^*\|_2 \|\boldsymbol{\eta}_t \odot \hat{\boldsymbol{m}}_t\|_2 \leq D\|\boldsymbol{\eta}_t \odot \hat{\boldsymbol{m}}_t\|_2$$

Using the bound on $\hat{m}_{t,i}$ from Lemma 8 and the definition of $\boldsymbol{\eta}_t$:

$$\|\boldsymbol{\eta}_t \odot \hat{\boldsymbol{m}}_t\|_2^2 = \sum_{i=1}^{d} \left( \alpha \frac{\hat{m}_{t,i}}{\sqrt{\hat{v}_{t,i}} + \epsilon} \right)^2 \leq \alpha^2 \sum_{i=1}^{d} \frac{(\hat{m}_{t,i})^2}{\hat{v}_{t,i}}$$

Applying Lemma 8, we have:

$$\sum_{t=1}^{T} \|\boldsymbol{\eta}_t \odot \hat{\boldsymbol{m}}_t\|_2^2 \leq \alpha^2 \sum_{i=1}^{d} \sum_{t=1}^{T} \frac{(\hat{m}_{t,i})^2}{\hat{v}_{t,i}} \leq \frac{\alpha^2 G_\infty^2}{(1-\beta_1)^2(1-\beta_2)} dT$$

Combining the above:

$$\sum_{t=1}^{T} (F(\hat{\boldsymbol{c}}_t) - F(\hat{\boldsymbol{c}}^*)) \leq \frac{D^2}{2\alpha(1-\beta_1)} \sum_{i=1}^{d} \sqrt{\hat{v}_{T,i}} + \frac{\alpha G_\infty^2}{(1-\beta_1)^2(1-\beta_2)} dT + \delta DT$$

Note that $(\phi^2 - 1) \leq 0$ since $\phi = \frac{1}{1+\alpha\lambda} < 1$.

Assuming that $\delta$ (the subgradient error) is small, the cumulative regret $\mathcal{R}(T)$ grows sublinearly with $T$, implying that the average regret $\mathcal{R}(T)/T$ converges to zero as $T \to \infty$.

This completes this proof.

$\square$

| Optimizer | IMLE | NID | CMAP | SPO | DBB |
|-----------|------|------|------|------|------|
| Adam | 0.37 | 0.39 | 0.39 | 0.26 | 0.52 |
| Adadelta | 0.40 | 0.57 | 0.37 | 0.27 | 0.69 |
| Adagrad | 0.67 | 0.43 | 0.67 | 0.47 | 0.47 |
| AdamW | 0.65 | 0.41 | 0.55 | 0.39 | 0.62 |
| RMSpp | 0.33 | 0.39 | 0.38 | 0.26 | 0.60 |
| AProx | 0.39 | 0.38 | 0.38 | 0.25 | 0.38 |

Table 4: Comparison of AProx optimizer with baseline algorithms on the invariant knapsack problem

## B  COMPARISON WITH EXISTING GRADIENT DESCENT OPTIMIZERS

In this section, we compare the Adaptive Proximal Gradient Optimizer (AProx) with common existing gradient descent optimizers, highlighting the key differences and advantages. The comparison focuses on how each optimizer handles gradient updates, learning rates, momentum, regularization, and their suitability for dealing with inexact gradients in the P+O framework.

Table 3: Comparison of AProx with Existing Optimizers

| Feature | SGD | RMSProp | Adam-type | AProx (Proposed) |
|---------|-----|---------|-----------|------------------|
| Gradient Update | $g_t = \nabla f(\hat{c}_t)$ | Same as SGD | Same as SGD | $g_t = \nabla f(\hat{c}_t) + g_t^{sur}$ |
| First Moment (Momentum) | Not used | Not used | $m_t = \beta_1 m_{t-1} + (1 - \beta_1)g_t$ | Same as Adam-type |
| Second Moment (Adaptive LR) | Not used | $v_t = \beta v_{t-1} + (1 - \beta)g_t^2$ | $v_t = \beta_2 v_{t-1} + (1 - \beta_2)g_t^2$ | $v_t$ same as Adam-type
$\hat{v}_t = \max(\hat{v}_{t-1}, v_t)$ |
| Bias Correction | Not applicable | Not used | $\hat{m}_t = \dfrac{m_t}{1 - \beta_1^t}$ | Same as Adam-type |
| Learning Rate | Fixed $\eta$ | $\eta_t = \dfrac{\eta}{\sqrt{v_t} + \epsilon}$ | $\eta_t = \alpha \dfrac{1}{\sqrt{\hat{v}_t} + \epsilon}$ | Same as Adam-type |
| Weight Decay | Not included | Not included | Varies (Adam: coupled, AdamW: decoupled) | Adaptive: $\hat{c}_{t+1} = \dfrac{1}{1 + \alpha\lambda}(\hat{c}_t - \eta_t \hat{m}_t)$ |
| Proximal Operator | Not included | Not included | Not included | Implicit via subgradient |
| Parameter Averaging | Not used | Not used | Not commonly used | $\hat{c}_{\text{avg},t} = \gamma \hat{c}_{\text{avg},t-1} + (1 - \gamma)\hat{c}_t$ |
| Handles Inexact Gradients | No | Partial (adaptive LR helps) | Partial (adaptive LR and momentum help) | Yes (designed for inexact gradients) |

**Adaptive Weight Decay**: AProx introduces an adaptive weight decay mechanism that dynamically scales the parameter updates, enhancing regularization and stability, especially important when dealing with inexact gradients. This is distinct from Adam-type optimizers where weight decay is either coupled with the learning rate (Adam) or decoupled but fixed (AdamW).

**Proximal Operator Integration**: AProx uniquely incorporates the proximal operator implicitly via the subgradient, making it suitable for optimization problems with nonsmooth regularization terms, which is not addressed by other optimizers.

**Handling Inexact Gradients**: AProx is specifically designed to handle inexact surrogate gradients inherent in the P+O framework, providing robustness and improved convergence. While RMSProp and Adam-type optimizers partially handle gradient noise due to adaptive learning rates and momentum, they are not tailored for the specific challenges posed by inexact surrogate gradients.

**Parameter Averaging**: AProx employs temporal parameter averaging to reduce sensitivity to noisy updates and improve generalization, a strategy not commonly used in other optimizers.

## C  COMPARISON WITH BASELINE OPTIMIZERS

| Optimizer | IMLE | NID | CMAP | SPO | DBB |
|-----------|------|-----|------|-----|-----|
| Adam | 0.17 | 0.19 | 0.18 | 0.15 | 0.20 |
| Adadelta | 0.08 | 0.20 | 0.15 | 0.04 | 0.19 |
| Adagrad | 0.20 | 0.21 | 0.20 | 0.20 | 0.20 |
| AdamW | 0.21 | 0.20 | 0.19 | 0.20 | 0.20 |
| RMSpp | 0.07 | 0.20 | 0.16 | 0.03 | 0.20 |
| AProx | 0.06 | 0.20 | 0.13 | 0.02 | 0.19 |

Table 5: Comparison of AProx optimizer with baseline algorithms on the portfolio problem

| Optimizer | IMLE | NID | CMAP | SPO | DBB |
|-----------|------|-----|------|-----|-----|
| Adam | 0.41 | 0.44 | 0.12 | 0.16 | 0.41 |
| Adadelta | 0.29 | 0.44 | 0.43 | 0.13 | 0.42 |
| Adagrad | 0.45 | 0.45 | 0.42 | 0.42 | 0.45 |
| AdamW | 0.43 | 0.45 | 0.13 | 0.40 | 0.44 |
| RMSpp | 0.42 | 0.44 | 0.12 | 0.13 | 0.41 |
| AProx | 0.17 | 0.31 | 0.10 | 0.12 | 0.37 |

Table 6: Comparison of AProx optimizer with baseline algorithms on the shortest path problem

