# OpenReview forum: "Adaptive Proximal Gradient Optimizer: Addressing Gradient Inexactness in Predict+Optimize Framework"
_ICLR.cc/2025/Conference — ICLR 2025 Conference Withdrawn Submission_

### Official Review · Reviewer_1hqx · 2024-10-31

**Soundness:** 1
**Presentation:** 1
**Contribution:** 1
**Rating:** 1
**Confidence:** 4

**Summary:**

The "predict then optimize” (P+O) framework is a two-step approach for decision-making in scenarios where optimization is dependent on uncertain data. First, a predictive model (e.g., neural network) estimates unknown parameters or outcomes (e.g., demand, prices, costs) based on historical or contextual data. Next, using these predictions as inputs, an optimization model (LP solver) determines the best decision according to an objective function (e.g., maximizing profit, minimizing cost) under given constraints.

The framework assumes that the true prediction parameters are available when training the predictive model. However, training with direct supervision on these parameters (e.g., with squared loss) ignores the downstream performance metric -- the regret. Unfortunately, regret is not differentiable, or the gradients are zero, which does not allow end-to-end training. Therefore, several methods proposed surrogate losses (like SPO+) or surrogate gradients (IMLE, CMAP, DBB, NID - as denoted in the paper).

The paper under review proposes an optimizer designed explicitly for the P+O framework that should utilize the surrogate gradients better than existing general optimizers. The method is inspired by proximal gradient descent and utilizes existing ingredients like momentum, adaptive lr, and smoothing.
The work claims some convergence guarantees in the convex case with a bound on convergence rates and empirically compares them to existing popular optimizers.

**Strengths:**

The paper tries to tackle an important problem in the popular P+O framework. Proving convergence is not straightforward, even in a simple setting with exact gradients.

**Weaknesses:**

The paper actually does not contain what promises. It seems that it actually only adds a squared loss on costs (prediction parameters) to an existing surrogate (or, equivalently, adds a gradient of squared loss to an existing surrogate gradient) in an obfuscated way and then basically uses a custom version of Adam optimizer and weight decay.

Specifically, It recalls the proximal gradient descent (Eq 8) but does not use it ("In practice, computing the proximal operator $prox_{\eta R}$ can be impractical in P+O problems. Instead, we utilize the existing surrogate gradient..."). Equation 9 then reveals that this surrogate with the gradient of $\ell^2$ norm is used.

Next, Theorem 1 is not proven correctly (and probably does not hold in this form):
- Equation 13 in the statement contains constant $\delta$, which is not quantified and is mentioned only in Eq. 5 in section 2.2. Here it requires that the bound is uniform in $\hat c$. It is mentioned that it is non-negligible. Indeed, the true gradient is always zero or undefined (since it is the gradient of the solution of an LP, i.e., of a piecewise constant function). Therefore $\delta=\sup_{\hat c}\|g(\hat c)\|$, where $g(\hat c)$ is the surrogate gradient.
- It is not clear, what $d_k$ is, since $\nabla R(\hat c_k)$ is not uniquely defined.
- l.834: The inequality does not hold (for instance, take $\eta$ close to $1/L$, then LHS is close to zero, but RHS is negative (close to $-\delta/L\|d_k\|$))
- L836: 'Higher order terms' $L\eta^2\delta^2$ cannot just 'be neglected' as there is no limit taken. Also, here it is incorrectly assumed that $\delta$ 'is small.'

The proof of Corollary 2 is wrong.
- Equation 44 does not imply that the sum converges. Consider for instance the sums $\sum_{k=1}^N 1/k -\sum_{k=1}^N 1/\sqrt k\le 0$, they both diverge.
(I skipped reading the proof of the next corollary)

The paper is full of incorrect or inexact statements:
- l.67 "The problem of gradient inexact caused by the agent function for P+O under the end-to-end framework has not been emphasized, and research is lacking." The above-mentioned papers are devoted to exactly this.
- l.36 "end-to-end approaches are also an emerging topic in the decision-making process."
2. Inexact Gradient Challenge in P+O Framework
- “The existence of errorbound can mislead the direction of descent, which will eventually lead to the problem of unstable or non-convergence of the training process.” The true gradient is always zero (or nonexistent); hence any informative descent direction will actually increase this delta. Therefore, there is no connection between this delta and some unstability or non-convergence (or maybe it is completely opposite, that for convergence, it is required delta to be large.)
3.  Adaptive Proximal Gradient Optimizer (AProx)
- l.165 "$R(\hat c)$ is not trivial". What does it mean?
- l.172 "The number 1/2 as coefficients of $f(\hat c)=\tfrac12|\hat c-c|^2$ is to avoid its excessive influence on the  gradient of the composite function." No, it is to avoid unnecessary constant 2 in the proximal gradient step.
- l.188 "This approach effectively integrates the proximal operator implicitly and allows us to proceed without its explicit computation." No. it just ignores the proximal map completely.
4. Theoretical Convergence Analysis
- l.240 "It is worth noting that Lemma 2 rests on the fact that R(·) is a convex function. In the solution approaches of P+O, most of the constructed surrogate functions can satisfy convexity." It is not true that "CMAP...involve convex functions" does not imply it is convex. "DBB uses linear interpolations..." (this is correct) but does not ensure that the result is convex (which is not, in general), similarly for IMLE.

It uses nonstandard or misleading terminology:
- l.59 "agent function" and "agent gradient" for surrogate loss/gradient.
- l.64 "discovergence"
- It often uses the term (and notation) 'gradient' for objects that are not, in fact, real gradients but surrogate ascent direction.
- "training rounds" (l.352 ), "step size" (l.367) and "calendar hours" (l.506) are used instead of epochs
- l. 80 "We ... give an inference on the rate of descent."
- l.74 "We propose the inexact surrogate gradient problem"
- l.77 "the optimizer, which is improved on the proximal gradient."
- l.156 "to address the inexact gradient challenge in Predict+Optimize (P+O) challenge"
- l.170 "we used the l2 paradigm term for the prediction error"

Experiments
- I am not able to understand the setting of the benchmarks in Tables 1, 3 and 5. It is claimed that "Table 1 shows the step size and training time per epoch required for convergence when training with several different regrets." I guess that "regrets" means "surrogates" and refers to one of the methods of IMLE, NID, CMAP, SPO, or DBB. However, it is not described how the statistics were calculated. Also, the std in the tables is so large that no conclusions can be drawn from it.
- I do not understand the experiment setup. The metric used (relative optimal gap) measures the performance of the trained model and not the optimizers. However, the training is not mentioned. Next, it is not clear how significant the results are. No statistical testing or even a std was reported.

Overall
- The paper proposes an enhancement to optimization within the P+O framework but lacks clarity and rigor in both theoretical claims and experimental evaluation.
- The main contribution—adding a squared loss to an existing surrogate with a custom optimizer—is presented obfuscated, does not bring any novel insights in the field, or does not help to understand the existing methods better.
- Theoretical issues arise, especially in Theorem 1 and Corollary 2, where the proofs contain major flaws.
- Misleading terminology and insufficient setup description and statistical analysis in experiments limit the work’s impact.

**Questions:**

I have no questions

---

> ### Author Response · Authors · 2024-11-28
>
> We are very grateful for the detailed and comprehensive feedback provided by Reviewer 3, especially regarding the theoretical aspects of our work. Your comments have greatly helped us to improve the rigour and clarity of our paper. The lead author is deeply grateful for these comprehensive comments and has made significant revisions to address most of the points.
>
> ---
>
> ### 1. Theorem 1 and Corollary 2 Issues
>
> > **Reviewer Comment:** "Theorem 1 is not proven correctly. Specific issues include undefined constant \(\delta\), unclear definitions for \(d_k\), and incorrect inequalities. The proof of Corollary 2 is wrong. Equation 44 does not imply convergence as claimed."
>
> **Response:**\
> We have thoroughly reworked the theoretical analysis. Instead of attempting to fix the original proof, we have adopted a new approach for proving Theorem 1, which provides a clearer and more robust foundation for our claims. Additionally, we delete Corollary 2, as it was redundant for new Theorem 1 in current manuscript. The revised proof for Theorem 1 with additional lemmas can be found in Appendix A.5.
>
> ---
>
> ### 2. Incorrect Statements and Terminology
>
> > **Reviewer Comment:** "The paper contains several inexact or misleading statements, such as the reference to 'gradient inaccuracy' in Line 67, which is already covered by the cited work."
>
> **Response:**\
> We have carefully reviewed the entire manuscript to address and revise any incorrect statements or misleading phrasing. Specifically, we have updated all references to gradient inaccuracies and ensured they are precise and properly contextualized within the literature. Additionally, we revised the discussion about the inexact gradient challenge to consistently use the term 'surrogate gradient' to indicate errors associated with the original inexact gradient concept(see Section 2.1).
>
> ---
>
> ### 3. Adaptive Proximal Gradient Optimizer (AProx)
>
> > **Reviewer Comment:** "The explanation regarding \(R(\hat{c})\) and the coefficients in \(f(\hat{c})\) lacks clarity. The statement regarding implicit integration of the proximal operator needs more justification."
>
> **Response:**\
> We have revised the explanation regarding the coefficient \(1/2\) in \(f(\hat{c})\). Our original intention aligns with the reviewer's comment, but we agree that the statement needed clarification. The updated manuscript now clearly explains that this coefficient helps to balance the regularization's influence, preventing it from overwhelming the gradient step (see Section 3.2).
>
> Furthermore, we have provided a formal justification for the implicit integration of the proximal operator by adding Lemma 1, which proves that the proximal gradient descent can be equivalently expressed as a subgradient method under certain conditions. This equivalence is explained in Section 3.1.
>
> ---
>
> ### 4. Theoretical Convergence Analysis
>
> > **Reviewer Comment:** "Lemma 2 relies on convexity assumptions that are not always valid."
>
> **Response:**\
> We have extended the theoretical convergence analysis by explicitly proving the convexity of three specific surrogate functions that are employed in our framework. These proofs are provided in Lemmas 2, 3, and 4, and they ensure the validity of our approach under practical conditions (see Section 4).
>
> ---
>
> ### 5. Nonstandard Terminology and Presentation
>
> > **Reviewer Comment:** "The paper uses nonstandard or misleading terminology, such as 'agent function' or 'discovergence'."
>
> **Response:**\
> We have reviewed the entire manuscript to ensure all terminology aligns with standard conventions. The terms "agent function" have been replaced with "surrogate gradient". We also corrected terms like "training rounds", "step size" and "calendar hours". Additionally, we addressed several other issues that had not been explicitly mentioned by the reviewers. We welcome any further suggestions to improve the clarity of our presentation.
>
> ---
>
> ### 6. Experimental Setup Clarification
>
> > **Reviewer Comment:** "The benchmark settings and experimental setup in Tables 1, 3, and 5 are unclear. Statistical analysis is missing."
>
> **Response:**\
> We have restructured the entire experimental section to provide greater clarity. Specifically, we have reorganized the descriptions of the benchmark settings, surrogate functions, and baselines in Section 6.1. We also added detailed explanations for Tables 1 and 2, specifying the experimental configurations and results. Regarding the large standard deviation issue, it is important to note that our convergence analysis was conducted across five different surrogate functions, leading to greater variability. This variability has been further explained in the revised manuscript (see Section 6.3). Additionally, we have provided an analysis of the significance of our experimental results by including the proportion of improvement in the optimal gap as shown in the ablation study, detailed in Table 2.

---

### Official Review · Reviewer_Gyq5 · 2024-10-31

**Soundness:** 1
**Presentation:** 3
**Contribution:** 2
**Rating:** 3
**Confidence:** 4

**Summary:**

This work proposes to use an adaptive proximal gradient optimizer in order to address issues arising in inexact gradient computations in predict+optimize works. The idea is to first add a smooth function $f$ to the regret $R$. Next, this work integrates adaptive learning rate, momentum, and parameter averaging in the minimization of $\Phi = f + R$.

**Strengths:**

1) The proposed work is interesting and aims to tackle a well-known issue arising in the non-differentiability of the loss function in Predict & Optimize.
2) The numerical experiments show promising results for the proposed approach.
3) The introduction and related works are well-written.

**Weaknesses:**

Major Comments:

1) The proof of the main theorem is incorrect. The inequality in Line 772 does not necessarily hold as a result of Line 765. The authors should revisit the proof and correct these details.
2) The paper claims that this paper uses a proximal update. However, the update is given by $\hat{c}_{k+1} = \hat{c}_k - \eta (\nabla f(\hat{c}) - g(\hat{c}))$, where  $g$ is an inexact gradient estimate of the non-smooth loss/regret term. The authors claim this is "implicitly a proximal update" in line 188, but this resembles a subgradient descent instead. It would benefit the paper if the authors could further elaborate on how this update relates to or a proximal update, or revise their claims if they cannot justify this connection.

3) The main theorem relies on $R(\hat{c}) = c^\top(z^\star(\hat{c}) - z^\star(c))$ being convex in $\hat{c}$. However, it is not obvious that the regret is convex. This paper would benefit if it either provides a proof of convexity for the regret function, or discuss the implications if this assumption does not hold and how it might affect the validity of the results.

Minor Comments:

1) The authors should update their references. For example, "Differentiation of Blackbox Combinatorial Solvers" is not cited properly, as it is already a published article.
2) The paper uses $\nabla R$ and $g$ interchangeably. However, $\nabla R$ is the true gradient of the regret (and assumes $R$ is differentiable), whereas $g$ is an approximation. The authors should update this in, e.g., Line 5 of the pseudocode and in line 259.
3) Line 052: "non-differentiate" -> "non-differentiable"
4) Line 068: "gradient inexact" -> "inexact gradients"
5) $R(\hat{c})$ is never explicitly written. It would make the paper more readable if the authors explicitly defined it in Section 2
6) Line 161: "introduces" -> "introduced"
7) Line 233 "approachfocuses" -> "approach focuses"

**Questions:**

1) How do you tune hyperparameters $\eta$, $\lambda$, $\beta_s, \beta_m, \beta_p$?
2) What does Table 7 show? Entries are denoted by "yes" and "no". There is no description in the caption. It is also not referenced in the main draft. The authors should remove unreferenced tables or reference them in the main draft.
3) Line 368 states that Table 1 shows training with "several different regrets". However, Table 1 only shows optimization algorithms and not regrets. Is this line referencing the wrong table?

---

> ### Author Response · Authors · 2024-11-28
>
> Thank you for your valuable and detailed comments. Your feedback on the correctness of the proof of the theorem, and the characterization of the proximal update have allowed us to revise some key issues to improve the accuracy of our theoretical arguments. Your feedback on minor problems such as symbol inconsistency and reference formatting also helped to improve the presentation of our work. We are deeply grateful for the time and expertise you invested in reviewing our manuscript.
>
> ---
>
> ### 1. Proof of Main Theorem (Line 772)
>
> > **Reviewer Comment:** "The proof of the main theorem is incorrect. The inequality in Line 772 does not necessarily hold as a result of Line 765. The authors should revisit the proof."
>
> **Response:**
> We have addressed the issue raised regarding the proof of the main theorem. Instead of modifying the original proof, we utilize a completely different approach to prove convergence. This new proof is supported by several additional lemmas, which strengthen the logical flow and ensure all assumptions are explicitly considered. For more details, please refer to Appendix A.5, where we outline each lemma and its role in the convergence proof. These supporting lemmas are crucial in verifying the conditions under which the inequality holds, thus establishing the validity of our results.
>
> ---
>
> ### 2. Proximal Update Clarification (Line 188)
>
> > **Reviewer Comment:** "The paper claims that this uses a proximal update, but it resembles subgradient descent instead."
>
> **Response:**
> We have clarified this point by providing Lemma 1, which shows that the subgradient can be seen as an implicit solution of the proximal gradient descent method. To make this clearer, we expanded our explanation in Section 3.1 and included the formal equivalence in Lemma 1. We hope these additions will help you differentiate the connection between our approach and the explicit proximal update method.
>
> ---
>
> ### 3. Convexity of the Regret Function
>
> > **Reviewer Comment:** "The main theorem relies on \( R(\hat{c}) \) being convex in \( \hat{c} \), but this is not obvious."
>
> **Response:**
> The convexity of \( R(\hat{c}) \) is indeed not immediately apparent, and we appreciate this point being highlighted. In response, we have extended our theoretical analysis to provide proofs of convexity for three specific surrogate functions \( 	tilde{R}(\hat{c}) \). These surrogate functions solve the non-differentiate challenge of P+O in different ways, of which the convexity has been rigorously demonstrated in Lemmas 2, 3, and 4 (see Section 4). These additional lemmas establish that regret minimization remains valid and convex under certain assumptions.
>
> ---
>
> ### Minor Comments
>
> 1. **References Update**
>    We have updated the reference for "Differentiation of Blackbox Combinatorial Solvers" to properly cite the published version, ensuring consistency throughout the manuscript.
>
> 2. **Gradient Notation**
>    We have thoroughly revised the gradient notation throughout the manuscript to make a clear distinction between \( \nabla R \), which represents the true gradient of the regret function, and \( \tilde{g} \), which is a surrogate gradient. This distinction is now explicitly defined in Section 3 (Page 4, Lines 162-170) to prevent confusion. If any ambiguity remains, please do not hesitate to inform us, and we will make further adjustments.
>
> 3. **Typo Corrections**
>    Due to extensive revisions to enhance overall quality, line numbers have changed. However, we have performed a comprehensive review of the manuscript, and we have ensured that all previously mentioned typographical errors have been corrected in the new version. We believe the quality of the writing has significantly improved, and we hope it now meets the expected standard.
>
> ---
>
> ### Questions
>
> 1. **Hyperparameter Tuning**
>    The hyperparameters \( \eta, \lambda, eta_s, eta_m, eta_p \) were tuned using a grid search methodology on a validation set. The specific ranges and values for each parameter are detailed in Theorem 1 and the experimental settings sections (see Section 6.1). These details provide insights into how the parameters were chosen to ensure robust performance across different benchmarks.
>
> 2. **Table 7 Description**
>    We have revised Section B to include a detailed explanation of Table 7, ensuring it is fully integrated into the main text and properly explained (see Section B). The caption for the table has also been updated to clarify that it presents hyperparameter settings and their impact on convergence. This makes the role of Table 7 more explicit and informative for readers.
>
> 3. **Correction to Table 1 Reference**
>    Regarding the reference to Table 1 and convergence epochs and training times, we have clarified that these metrics are averaged across different regret functions, showing the mean and standard deviation. The caption for Table 1 has been updated accordingly, and we provided further explanations in the experiment settings.

---

### Official Review · Reviewer_nPn2 · 2024-11-07

**Soundness:** 1
**Presentation:** 1
**Contribution:** 1
**Rating:** 1
**Confidence:** 3

**Summary:**

This paper addresses the predict and optimize framework, which utilises learning algorithms to predict parameters for optimization problems in an end to end fashion. Unforutnately, incorporating the optimization stage into the problem results introduces nonsmoothness into the objective. The paper addresses this issue by utilizing a proximal framework. The authors analyse the theoretical convergence and practical performance of their algorithm.

**Strengths:**

- The topic of the paper is interesting.
- The paper provides a review of related works.

**Weaknesses:**

There is a major error in the proof of theorem 1: on line 772 a lower bound is incorrectly combined with an upper bound. Since this is the basis for the main convergence result, this error compromises the theoretical results presented.

I believe there is another error in the proof of corollary 2 (883-886). The sequence $||d_k|| $ need not converge to zero. For example, consider, $||d_k|| = (\delta\eta)/c$ which satisfies (44) but clearly does not converge to zero.

I also believe that there are major flaws in the specification of Algorithm 1. For example on line 3 the $ \hat{c_k} = \hat{c}(\theta_k) $, while in line 10 $ {\hat{c}}_{k+1} $ is computed as an update sequence based on the gradient. On line 12 a set of smoothed $ \tilde{\hat{c}}_k $ are computed but not utilised (so far as I can tell).

Another concern I have with is the smoothing function selected by the authors. To compute the gradient of $f(\hat{c}) = \frac{1}{2} || \hat{c} - c||^2$ with respect to $\hat{c}$ requires knowledge of the "true cost parameters", which precludes practical implementation.

Additionally, paper is significantly hindered by the quality of the writing with many awkward and confusing sentences, confusing notation and typos. The paper is difficult to follow due to theses issues. For example, in section 2.1 equation (3) is stated with no relation to the previous paragraph and (4) is stated with no discussion. There are issues like this in almost every section of the paper.

**Questions:**

See the weaknesses section. If the authors can clarify the theoretical concerns and substantially improve the quality of the text, I will be happy to take a second look at the paper.

---

> ### Author Response · Authors · 2024-11-28
>
> We sincerely appreciate your thorough review and detailed feedback. Your comments on the proof of Theorem 1, the implementation of Algorithm 1, the practical significance of the smoothing function, and the overall quality of the manuscript were particularly insightful. Addressing these issues has helped us to strengthen both the theoretical and practical aspects of our work.
>
> ---
>
> ### 1. Proof of Theorem 1 and Corollary 2
>
> > **Reviewer Comment:** "There is a major error in the proof of Theorem 1: on line 772, a lower bound is incorrectly combined with an upper bound. Since this is the basis for the main convergence result, this error compromises the theoretical results presented."
>
> > **Reviewer Comment:** "I believe there is another error in the proof of Corollary 2. The sequence need not converge to zero."
>
> **Response:**
> We agree that there was an error in the proof of Theorem 1 where a lower bound was incorrectly combined with an upper bound. This oversight undermined the validity of the convergence result. To address this, we have thoroughly revised the proof of Theorem 1. We adopted a new convergence proof strategy that correctly handles the combination of bounds. Specifically, we restructured the proof to ensure that all inequalities are applied appropriately and that each step is rigorously justified. This revised approach avoids the previous error and strengthens the theoretical foundation of our convergence result.
> The updated proofs can be found in Section 4 of the revised manuscript, with detailed explanations and supporting lemmas provided in Appendix A.5. We believe these revisions not only correct the errors but also enhance the overall rigor of our theoretical results.
>
> ---
>
> ### 2. Algorithm 1 Specification and Use of Smoothed Parameters
>
> > **Reviewer Comment:** "There are major flaws in the specification of Algorithm 1. On line 12, a set of smoothed are computed but not utilized."
>
> **Response:**
>
> We have updated Algorithm 1 to modify the gradient update rule to ensure that the smoothed parameters are explicitly utilized. The role of the smoothed parameters was clarified, ensuring that their effect is fully accounted for in the subsequent updates to provide adaptive regularization. This is reflected in the parameter averaging mechanism added on Page 5, Lines 13-15, which now clearly demonstrates how the smoothed influences the model parameters to improve stability during optimization.
>
> ---
>
> ### 3. Smoothing Function and Practical Implementation
>
> > **Reviewer Comment:** "Another concern is the smoothing function selected by the authors. To compute the gradient of with respect to requires knowledge of the 'true cost parameters', which precludes practical implementation."
>
> **Response:**
> Your consideration regarding the requirement for true cost parameters is reasonable. However, a fundamental assumption of this work is that the true cost parameters \(c\) can be accessed, which is based on the premise that a predictive model in machine learning can be used under constructed training datasets. Even for other methods, if one intends to evaluate the performance of a predictive model, access to true cost parameters is essential. Otherwise, the prediction would be an open-loop forward pass without meaningful feedback.
>
> ---
>
> ### 4. Presentation and Writing Quality
>
> > **Reviewer Comment:** "The paper contains many awkward and confusing sentences, confusing notation, and typos. For example, in Section 2.1, Equation (3) is stated without relation to the previous paragraph, and Equation (4) is presented with no discussion."
>
> **Response:**
> In addition to the specific revisions mentioned, we have performed an extensive revision throughout the manuscript to enhance the overall readability and improve the flow of arguments. We have worked on ensuring that all terminology, notation, and transitions are consistent and clear. If there are still areas that remain unclear, we are more than happy to address them to the best of our ability. Your feedback is valuable, and we are committed to further refining the work if necessary.

---

### Author Response · Authors · 2024-11-28
**Overall Response to Reviewers**

We are very grateful for the insightful suggestions provided by every reviewer, and we believe that these comments have greatly contributed to enhancing the quality of our work. Additionally, we appreciate the program committee for extending the timeline, which allowed us time to make comprehensive revisions to our paper. Below, we provide a summary of the common issues raised by the reviewers, followed by a detailed explanation of how we addressed them.

**1. Theoretical Proof Issues (Theorem 1 and Corollary 2 - Reviewers nPn2, Gyq5, 1hqx):**
We acknowledge the errors identified in Theorem 1 and Corollary 2 of the initial manuscript. In response, we have entirely revised the theoretical analysis section, adopting a different approach for the proof of convergence and supportive lemmas (see Section 4 and Appendix A).

**2. Algorithm 1 Clarifications (Reviewer nPn2):**
Significant updates have been made to Algorithm 1 to enhance clarity and correctness. Specifically, the role of smoothed parameters (line 12) in providing adaptive weight decay was not clearly articulated. We have now included a detailed explanation to clarify their role in smoothing gradient fluctuations during training (see Page 5, lines 1-15).

**3. Proximal Update Method (Reviewer Gyq5):**
The original description of the proximal update was misleading, as it implied the direct use of a proximal operator. Instead, we approximate this using subgradients, effectively forming an implicit proximal-like method to reduce computational complexity. We revised the explanation in the manuscript accordingly to clarify the implicit nature of this approach and its relationship to end-to-end training (see Page 4, lines 162-174, and Section 4).

**4. Presentation and Writing Quality (Reviewer nPn2):**
We agree that several sections lacked consistency in terminology and clarity of presentation. We have revised the manuscript to ensure that terms like "gradient," "agent function," and "surrogate" are used consistently throughout. We have also restructured Section 2.1 to establish a logical progression from Equation (3) to (4), improving the overall flow of the discussion.

**5. Experimental Design and Metrics Clarification (Reviewers Gyq5, 1hqx):**
We have addressed the concerns regarding experimental clarity by reorganizing Section 6. This includes clarifying the experimental settings for Tables 1, 3, and 5, along with enhanced descriptions of how different surrogate methods contribute to the observed regret. Statistical tests have been conducted to improve the reliability of our findings.

**6. Minor Comments and Typos (Reviewer Gyq5):**
All noted minor issues have been corrected. These include fixing typos, clarifying ambiguous notations, and correcting incorrect references (e.g., the citation for "Differentiation of Blackbox Combinatorial Solvers"). We also ensured consistent use of symbols, such as ∇R and g, throughout the text.

---

### Note · Authors · 2025-05-06

I have read and agree with the venue's withdrawal policy on behalf of myself and my co-authors.

---

### Meta-Review · Area_Chair_hgNt · 2024-12-08

**Metareview:**

Unfortunately, the theoretical derivations of some of the main results in the paper appear to be incorrect. In their rebuttal, the authors claim to have addressed these issues by entirely adopting a different approach to redo the theory. However, this does not provide the reviewers with adequate time to reassess the entire theoretical derivation of the paper. Additionally, the paper suffers from poor quality in presentation and writing. As it stands, the paper may require more substantial changes and a subsequent reassessment beyond what the rebuttal process allows.

**Additional Comments On Reviewer Discussion:**

The major issues with the paper are the inaccurate theoretical derivations and the poor quality of presentation and writing.

---

### Decision · Program_Chairs · 2025-01-22

Reject